# Tackling Biased Evaluators in Dueling Bandits

**Ming Tang**
Dept. of Computer Science and Engineering
Southern Univ. of Science and Technology
Shenzhen, Guangdong, China
`tangm3@sustech.edu.cn`

**Yuxuan Zhou**
Dept. of Mathematics
Southern Univ. of Science and Technology
Shenzhen, Guangdong, China
`zhouyx8@mail.sustech.edu.cn`

**Chao Huang**[*]
School of Computing
Montclair State University
Montclair, New Jersey, USA
`huangch@montclair.edu`

## Abstract

In dueling bandits, an agent explores and exploits choices (i.e., arms) by learning from their stochastic feedback in the form of relative preferences. Prior related studies focused on unbiased feedback. In practice, however, the feedback provided by evaluators can be biased. For example, human users are likely to provide biased evaluation towards large language models due to their heterogeneous background. In this work, we aim to minimize the regret in dueling bandits considering evaluators' biased feedback. We begin with a benchmark case where evaluators' bias information is known. Solving the known-bias case is nontrivial, because the bias cannot be easily decoupled from the feedback. We overcome this challenge and propose an unbiased arm performance estimator and a bias-sensitive dueling bandits algorithm. We manage to analyze the regret, dealing with the complex form of the estimator, and show that the feedback either matching or opposing the ground-truth reduces the regret. Then, we study the case where evaluators' bias information is unknown. The associated estimator can hardly be solved in closed-form due to the non-convexity of the estimator solving problem. We address this challenge and propose an extended bias-sensitive algorithm by incorporating block coordinate descent. This algorithm is proven to achieve the same order of regret (as in the known bias case) with a bounded error. Experiments show that when compared with baselines, our algorithms reduces the regret by up to 86.9%.

## 1 Introduction

### 1.1 Motivation and Background

Multi-armed bandit (MAB) [1] is a widely used approach for online learning. It explores and exploits a given set of choices (i.e., arms) to minimize a long-term regret. In standard MAB, the reward of the selected arm is commonly represented by a real number, e.g., if pulling an arm of a slot machine returns 5 dollars, then the reward can be represented by 5. As a result, the exploration and exploitation decisions can be made based on these real-valued reward feedback. However, in many practical systems, the real-valued reward feedback is unavailable. For example, consider a company that aims at providing its users with high-quality user experience for question answering tasks by selecting

---

[*]Corresponding author

from various large language models (LLMs), e.g., GPT-4 [2], where these LLMs can be thought of arms. Unlike prediction and classification, the output of an LLM is usually paragraphs that are intrinsically subjective. Their ground-truth quality is hard to measure or may not even exist. This makes it difficult to use a real-valued reward to represent the quality of and select an LLM [3, 4].

To address the unavailability of real-valued reward feedback, existing studies (e.g., [5–14]) evaluated arms based on qualitative comparison between a pair of arms, which are referred to as *dueling bandits*. In these approaches, an agent selects two arms in each round for comparison. The agent then observes the qualitative comparison result between the two arms, based on which the agent makes exploration and exploitation decisions. Interested readers can refer to [15] for a comprehensive survey.

Although these studies (e.g., [9, 15]) addressed the lack of real-valued reward feedback, they did not consider an important scenario where the **feedback is provided by biased evaluators**. For example, [16] suggested that the LLM selection of a company (which serves as an agent) should be based on its users' feedback. However, the users (who serve as evaluators) are humans. Their feedback may be biased due to various factors, e.g., users' expertise or demographic background. Biased feedback can significantly degrade the performance of conventional dueling bandits approaches and increase the long-term regret. We empirically show that the presence of biased evaluators increases the regret of baselines by an average of $8.44$ folds (see Appendix K.8).

Some recent studies (e.g., [17, 18]) considered biased feedback in conventional MAB settings. However, those approaches are not applicable in dueling bandits due to a lack of real-valued rewards. Other studies (e.g., [19–21]) considered pairwise assessment with bias in mobile crowdsourcing, while their goal is to find the best choice or the ranking of choices without considering the long-term exploration-exploitation tradeoff. Thus, their algorithms and analytical frameworks are not applicable to dueling bandits. While adversarial dueling bandits (e.g., [22]) emphasized time-varying winning probability matrices of arms, they do not consider evaluator-specific biased feedback.

## 1.2 Solution and Approach

In this work, we take into account the feedback provided by biased evaluators and propose bias-sensitive upper confidence bound (UCB) algorithms with performance guarantee. Our proposed approach for addressing biased evaluators can be readily extended to other dueling bandits algorithms (e.g., relative confidence [10], relative UCB [11], double Thompson sampling [12]) and improve their performance (see Section 5). Specifically, we aim to answer the following questions:

Q1 How can we design an unbiased estimator for arm performance and a low-regret dueling bandits algorithm in the presence of evaluators' bias?

Q2 What is the performance guarantee of our algorithm?

Answering Q1 is challenging. (i) The bias of evaluators is usually unknown *a priori*. Thus, the algorithm design requires a joint estimation of the evaluators' bias and the winning probability of arms, which makes the corresponding estimator solving problem non-convex. (ii) Even when the bias of evaluators is known, such a design is non-trivial. An intuitive solution is to directly decouple the bias from the observed feedback by equation transformation. However, this is proven to induce an unbounded regret. We overcome the challenge by transforming the estimator design problem into a convex optimization problem and theoretically derive an unbiased estimator and its confidence radius.

Answering Q2 is non-trivial, as the determined estimator and confidence radius from Q1 involve evaluators' heterogeneous bias levels, which makes the regret analysis for conventional dueling bandits inapplicable. We overcome this challenge by applying equation transformation and introducing auxiliary inequalities to support the regret analysis and derive the regret of our proposed algorithms.

Our main contributions are listed as follows:

- To the best of our knowledge, this is the first attempt that considers biased evaluators in dueling bandits. Our approach is applicable to general arm performance models with deterministic winning probability and general bias models that model feedback with conditional probability. Meanwhile, it can be incorporated into existing dueling bandits approaches to reduce their regrets under the presence of evaluators' bias.

- We begin with the case where each evaluators' bias level is known. We overcome challenge Q1-(ii) and propose a bias-sensitive UCB algorithm. To address Q2, we theoretically derive the long-term regret. Analytical results show that our proposed algorithm achieves

a sublinear regret, which is of the same order to those in conventional UCB algorithms of dueling bandits.

- We further study the case where each evaluators' bias level is unknown. We overcome challenge Q1-(i) by decoupling evaluators' bias from arm performance estimation when initializing estimators and incorporating block coordinate descent (BCD) [23]. We propose an extended bias-sensitive UCB algorithm, and prove that this extended algorithm achieves the same order of regret as in the known bias case with a bounded error.

- Experiments show that when compared with five baselines, our algorithms reduces the regret by up to 86.9%. The reduction is more significant when the bias levels among evaluators are more heterogeneous. Meanwhile, our estimator can be incorporated into baselines and reduces their regrets by up to 75.9%.

## 2 System Setup

We consider an agent and a set of $M$ evaluators $\mathcal{M} = \{1, 2, ..., M\}$ whose feedback can be biased. There are a total of $K$ arms, denoted by set $\mathcal{K} = \{1, 2, ..., K\}$. In each time slot $t \in \mathcal{T} = \{1, 2, ..., T\}$, an arbitrary evaluator arrives. The agent selects two arms for the evaluator. We consider a setting where the evaluator evaluates the selected arms and, at the same time, provides pairwise comparison feedback for the arms. Consider LLM evaluation as an example. A company (agent) selects two LLMs (arms) to serve its users (evaluators). The users observe the inference output of the LLMs and provide pairwise comparison feedback for the two LLMs. The goal of the agent is to minimize the long-term regret of the selected arms (roughly speaking, maximize the chance that the best arm is selected) based on the evaluators' feedback.[2]

**Arm Model:** We consider a stochastic setting where an arm outperforms another arm with certain probability [15]. This probability is associated with the ground-truth performance of arms and cannot be observed directly. Let $o_i \succ o_j$ denote an observation that arm $i \in \mathcal{K}$ outperforms arm $j \in \mathcal{K}$, and let $\Pr(o_i \succ o_j)$ denote the probability that arm $i$ outperforms $j$. For ease of presentation, we denote

$$p_{ij} \triangleq \Pr(o_i \succ o_j). \tag{1}$$

We assume $\Pr(o_i \succ o_j) + \Pr(o_i \prec o_j) = 1$, and do not consider the case where comparing $o_i$ and $o_j$ leads to tie. As suggested by [24], ties can be handled by giving "half a point" to both arms, reducing the problem to a tie-free case. Note that probability model in (1) generalizes various models as special cases, e.g., Bradley-Terry (BT) model [21] and Logistic model [25].

In dueling bandits, a Condorcet winner (i.e., an arm $i$ with $p_{ij} > 1/2$ for all $j \in \mathcal{K} \setminus \{i\}$) may not exist [15]. As in many related works (e.g., [13, 14]), we define the best arm using *Borda score*:

$$\theta_i \triangleq \frac{1}{K-1} \sum_{j \in \mathcal{K} \setminus \{i\}} p_{ij}. \tag{2}$$

Intuitively, a larger $\theta_i$ implies a higher probability that arm $i$ beats other arms on average. This metric is suitable. For example, in LLM evaluation, a higher winning probability implies a higher chance that users are satisfied with the inference results of the LLM. We consider *Borda winner* [13, 14]:[3]

**Definition 1** (Borda Winner). *The best arm $i^*$ is the arm with the highest Borda score, i.e., $i^* = \arg \max_{i \in \mathcal{K}} \theta_i$.*

**Evaluator Bias Model:** We use $o_i \succ_m o_j$ to denote the case where evaluator $m \in \mathcal{M}$ provides a feedback claiming that arm $i$ outperforms arm $j$. Note that $o_i \succ_m o_j$ and $o_i \succ o_j$ *may not match* due to the bias of evaluator $m$. There are various types of evaluators' bias. In this work, we follow mobile crowdsourcing studies (e.g., [21]) and introduce a coefficient $\eta_m$ to characterize the probability that evaluator $m$ reveals a feedback that matches the ground-truth comparison result:

$$\eta_m \triangleq \Pr(o_i \succ_m o_j \mid o_i \succ o_j). \tag{3}$$

---

[2]Although we use LLM as a motivating example, this work focuses on a general dueling bandits scenario without targeting any particular application. To adapt it to the LLM setting, additional factors such as contextual information would need to be incorporated into the arm selection and comparison process.

[3]Despite the rationale of using Borda winner, it may sometimes be inconsistent with the Condorcet winner (if it exists). Thus, if finding the Condorcet winner is the primary goal, although our algorithms can still lead to superior performance (see Section 5), the theoretical analyses in this work may no longer be applicable.

That is, given the fact that $o_i \succ o_j$, evaluator $m$ with bias $\eta_m$ claims $o_i \succ_m o_j$ with probability $\eta_m$. Note that $\Pr(o_i \succ_m o_j \mid o_i \succ o_j) + \Pr(o_i \prec_m o_j \mid o_i \succ o_j) = 1$. Similarly, we exclude the case where the evaluator reports no difference between arms. If this case happens, the evaluator can randomize among the arms with equal probability and provides feedback. The bias model in (3) can characterize various types of bias, such as ambiguity in perception and comparison [26] and diverse roles of the evaluators [21]. Consider bias resulting from diverse roles as an example. If $\eta_m = 1$, then evaluator $m$ is a *perfect evaluator*. If $\eta_m = 0.5$, then evaluator $m$ is a *spammer* who provides random feedback. If $\eta_m = 0$, then evaluator $m$ is an *attacker* which aims to worsen the choice of the agent and always provides opposite feedback.

Based on (3), the probability that evaluator $m$ claims arm $i$ outperforms arm $j$ is given by

$$p_{ij}^m \triangleq \Pr(o_i \succ_m o_j) = \eta_m p_{ij} + (1 - \eta_m) p_{ji}. \tag{4}$$

**Arm Selection and Regret:** In time slot $t \in \mathcal{T}$, an evaluator arrives, and let $m_t \in \mathcal{M}$ denote this evaluator. The agent selects two arms $x_1(t) \in \mathcal{K}$ and $x_2(t) \in \mathcal{K}$ for the evaluator using a dueling bandits algorithm (to be proposed in Sections 3 and 4). Let $\boldsymbol{x}(t) \triangleq \{x_1(t), x_2(t)\}$. Note that $x_1(t) \neq x_2(t)$ must hold before algorithm convergence; otherwise, no comparison between arms is performed and hence there is no exploration in time $t$. After evaluator $m_t$ evaluates both chosen models $x_1(t)$ and $x_2(t)$, it sends a binary feedback to the agent, i.e., either $o_{x_1(t)} \succ_{m_t} o_{x_2(t)}$ or $o_{x_2(t)} \succ_{m_t} o_{x_1(t)}$. The binary feedback is commonly considered in dueling bandits [15] and is suitable for the scenario that lacks real-valued reward feedback from evaluators. Recall that in the LLM example, it is easy for users to judge which output from the two LLMs is better, while it is difficult for them to give real-valued score for the outputs of LLMs.

In this work, we focus on both *average regret* and *weak regret*, which are commonly considered regrets in dueling bandits [1]. The average regret $\mathbf{RegA}(\boldsymbol{x}(t))$ [10, 12] and weak regret $\mathbf{RegW}(\boldsymbol{x}(t))$ [27] are defined as the average and maximum Borda score among the two selected arms, respectively:

$$\mathbf{RegA}(\boldsymbol{x}(t)) = \theta_{i^*} - (\theta_{x_1(t)} + \theta_{x_2(t)})/2, \tag{5}$$

$$\mathbf{RegW}(\boldsymbol{x}(t)) = \theta_{i^*} - \max\{\theta_{x_1(t)}, \theta_{x_2(t)}\}. \tag{6}$$

For example, average regret refers to the case where a user retrieves information from the inference outputs of both LLMs. Weak regret refers to the case where a user is satisfied as long as one of the LLMs provides satisfactory output. Since *all of our algorithms and theoretical results apply to both average regret and weak regret,* we use $\mathbf{Reg}(\boldsymbol{x}(t))$ to denote them.

The goal is to minimize the long-term round-average regret:

$$\min_{[\boldsymbol{x}(t)]_{t=1}^T} \frac{1}{T} \sum_{t=1}^T \mathbb{E}[\mathbf{Reg}(\boldsymbol{x}(t))]. \tag{7}$$

We solve problem (7) for both known and unknown bias cases in Sections 3 and 4, respectively.

# 3 Known Bias Case

In this section, we start with the benchmark case where the evaluators' bias $\eta_m$ is known. In practical systems, the bias could be obtained by running pre-evaluation tests, e.g., in the LLM example, the company may estimate user bias through offering queries whose ground-truth answers are known. We consider the setting where the set of available bias is finite. That is, $\eta_m \in \mathcal{B} \triangleq \{\eta_1^A, \eta_2^A, ..., \eta_B^A\}$ for all $m \in \mathcal{M}$, where $B = |\mathcal{B}|$ and the superscript A is short for "available". As long as the number of evaluators is finite, this assumption on finite bias set holds.

We build our algorithm based on UCB. Despite this, our ideas for addressing biased evaluators can be incorporated into various baselines to reduce their regret (see Appendix K.1). Note that even for the known bias case, designing the algorithm is challenging. This is because when estimating the pairwise winning probability of arms, the bias cannot be easily decoupled from the observed feedback provided by evaluators. Meanwhile, the complex form of the winning probability estimator makes deriving the associated confidence radius and analyzing round-average regret further challenging.

## 3.1 Bias-Sensitive UCB Algorithm

We first present the unbiased estimation of pairwise winning probability of arms and confidence radius calculation respectively. Then, we show the algorithm details.

**1) Unbiased Arm Performance Estimation:** We aim to design an unbiased estimator of winning probability matrix $\boldsymbol{p} \triangleq (p_{ij}, i, j \in \mathcal{K})$,[4] which will be incorporated into our bias-sensitive UCB algorithm. Note that it is possible to obtain an unbiased estimator by transforming the problem into conventional dueling bandits via decoupling the bias in (4). However, such an estimator is sensitive to the feedback of spammers, which can lead to an infinite round-average regret (see Appendix A). To deal with this challenge, we first transform the estimator design problem into an optimization problem. Then, we solve the problem to obtain the estimator.

Let $N_{ij}^b(t)$ denote the number of feedback claiming $o_i \succ_m o_j$, and its evaluator has a bias $\eta_m = \eta_b^{\mathrm{A}}$. Let $\mathcal{B}_{ij}(t) \subseteq \mathcal{B}$ denote the set of bias index $b$ such that $N_{ij}^b(t) + N_{ji}^b(t) > 0$. Designing an estimator $\hat{\boldsymbol{p}}(t) = (\hat{p}_{ij}(t), i, j \in \mathcal{K})$ is equivalent to finding the optimal estimator $\hat{\boldsymbol{p}}(t)$ that minimizes the difference between the estimated value of $p_{ij}$ using the estimator and the approximate value $\bar{p}_{ij}^b(t) \triangleq N_{ij}^b(t)/(N_{ij}^b(t) + N_{ji}^b(t))$. That is, $\hat{\boldsymbol{p}}(t)$ minimizes the following problem:

$$\min_{\boldsymbol{p}} \sum_{i,j \in \mathcal{K}, b \in \mathcal{B}_{ij}(t)} (\eta_b^{\mathrm{A}} p_{ij} + (1 - \eta_b^{\mathrm{A}}) p_{ji} - \bar{p}_{ij}^b(t))^2. \tag{8}$$

Problem (8) contains $K^2 \times M$ terms, each corresponding to exactly one decision variable $p_{ij}$. Thus, problem (8) can be equivalently transformed to a set of sub-problems of $\hat{p}_{ij}(t)$:

$$\hat{p}_{ij}(t) = \arg \min_{p_{ij}} \|\mathbf{w}_{ij} p_{ij} + \mathbf{c}_{ij}\|^2, \tag{9}$$

where $\mathbf{w}_{ij} \triangleq (2\eta_b^{\mathrm{A}} - 1, b \in \mathcal{B}_{ij}(t))$, $\mathbf{c}_{ij} \triangleq (1 - \eta_b^{\mathrm{A}} - \bar{p}_{ij}^b(t), b \in \mathcal{B}_{ij}(t))$. Based on Karush-Kuhn-Tucker (KKT) conditions, the optimal solution to problem (9) satisfies $(\mathbf{w}_{ij}^\top \mathbf{w}_{ij}) \hat{p}_{ij}(t) = -\mathbf{w}_{ij}^\top \mathbf{c}_{ij}$. This results in the following unbiased estimator, with proof in Appendix B.

**Lemma 1** (Arm Performance Estimator). *After time slot $t$, the pairwise winning probability $p_{ij}$ in (1) is estimated by*

$$\hat{p}_{ij}(t) = \frac{\sum_{b \in \mathcal{B}_{ij}(t)} (2\eta_b^{\mathrm{A}} - 1) \left( \bar{p}_{ij}^b(t) - (1 - \eta_b^{\mathrm{A}}) \right)}{\sum_{b \in \mathcal{B}_{ij}(t)} (2\eta_b^{\mathrm{A}} - 1)^2}. \tag{10}$$

*This estimator is unbiased, i.e., $\mathbb{E}[\hat{p}_{ij}(t)] = p_{ij}$. Based on (10), if an evaluator tends to be a spammer (i.e., $\eta_m$ is closer to $0.5$), a lower weight is assigned to the evaluator's feedback.*

**2) Confidence Radius Calculation:** We now derive the confidence radius of the estimator in Lemma 1. This analysis is more challenging than that in conventional dueling bandits, because the estimator is in the form of a weighted sum of the feedback statistics of evaluators considering their bias. The involved sum, weighting, and shift operations require additional mathematical transformation to solve the confidence radius based on Hoeffding inequality. The proof is given in Appendix C.

**Definition 2** (Confidence Radius). *We define the confidence radius as $\Pr(|\hat{p}_{ij}(t) - p_{ij}| \le r_{ij}(t)) \ge 1 - 2/t^{2\alpha}$. That is, $r_{ij}(t)$ is a one-dimensional bound such that $|\hat{p}_{ij}(t) - p_{ij}| \le r_{ij}(t)$ occurs with a probability no smaller than $1 - 2/t^{2\alpha}$, where parameter $\alpha > 0$ controls the required probability.*

**Proposition 1** (Confidence Radius). *The confidence radius $r_{ij}(t)$ in Definition 2 is determined by*

$$r_{ij}(t) = \frac{\sum_{b \in \mathcal{B}_{ij}(t)} |2\eta_b^{\mathrm{A}} - 1| \sqrt{\frac{\alpha \log(t)}{(N_{ij}^b(t) + N_{ji}^b(t))}}}{\sum_{b \in \mathcal{B}_{ij}(t)} (2\eta_b^{\mathrm{A}} - 1)^2}, \tag{11}$$

*where $\log(t)$ is of natural base.*

**3) Algorithm Details:** We now present the bias-sensitive UCB algorithm. The pseudocode is provided in Algorithm 1 of Appendix D. The algorithm iterates for $T$ rounds or until convergence. At the beginning of each time slot $t$, the agent updates $\hat{p}_{ij}(t-1)$ using (10) and $r_{ij}(t-1)$ using (11). Then, it computes the upper confidence bound estimation of probability $p_{ij}$:

$$\mathrm{UCB}_{ij}(t) = [\hat{p}_{ij}(t - 1) + r_{ij}(t - 1)]^-, \tag{12}$$

where $[\cdot]^- \triangleq \min\{\cdot, \overline{\mathrm{UCB}}\}$. With this operator $[\cdot]^-$, the agent tends to randomly explore if all arms are under-explored. We set $\overline{\mathrm{UCB}} = 1$ in the experiments [13]. In (12), if $\hat{p}_{ij}(t-1)$ is larger than $1/2$, then arm $i$ is likely to outperform arm $j$ based on the historical observation, indicating a higher

---

[4]The term "bias" in "unbiased estimator" differs from that in "bias of evaluator". "Unbiased estimator" implies that the expected value of the estimator equals the true value being estimated.

reward through exploiting model $i$. If $r_{ij}(t-1)$ is larger, then the uncertainty regarding arms $i$ and $j$ is higher, indicating stronger need to compare arms $i$ and $j$ in the following time slot.

After that, the agents computes the UCB estimation of Borda score:

$$\text{UCB}_i(t) = \frac{1}{K-1} \sum_{j \in \mathcal{K} \setminus \{i\}} \text{UCB}_{ij}(t). \tag{13}$$

Finally, the agent selects the two arms $x_1(t)$ and $x_2(t)$ with the maximum values of $\text{UCB}_i(t)$:

$$\max_{\boldsymbol{x}(t)} \ \text{UCB}_{x_1(t)}(t) + \text{UCB}_{x_2(t)}(t). \tag{14}$$

Different from some existing works (e.g., [11]) in dueling bandits that choose the best arm (e.g., with the highest UCB) and its "strongest competitor", our algorithm chooses the best and second best arms (e.g., with the highest and second highest UCB values) for analytical simplicity. In Appendix E, we empirically show that replacing the second arm with the "strongest competitor" may degrade the performance, especially when the number of arms is large or when a Condorcet winner does not exist.

## 3.2 Regret Analysis

We now bound the round-average regret of the proposed algorithm. The proof is given in Appendix F. The proof path follows [13], while it is more difficult due to the complex form of the estimator and confidence radius. Note that we essentially derive the bound for average regret. This bound is also applicable to weak regret by relaxing it to average regret in the proof (see Appendix F).

**Theorem 1** (Regret of Bias-Sensitive UCB Algorithm). *The bias-sensitive UCB algorithm with $T$ rounds has a round-average regret of*

$$\frac{1}{T} \sum_{t=1}^{T} \boldsymbol{Reg}(\boldsymbol{x}(t)) \leq \frac{\overline{UCB}\,(K(K-1)+2H)}{T}$$

$$+ \frac{2\overline{UCB}\sqrt{\alpha \log(T)}}{\Gamma} \left( \frac{H + B^2 \log(BT^{2\alpha})}{T} + \sqrt{\frac{2BK}{K-1}} \cdot \frac{1}{\sqrt{T}} \right). \tag{15}$$

*where $H = \sum_{t=K(K-1)/2+1}^{\infty} t^{-2\alpha}$, and $\Gamma \triangleq \sum_{b \in \mathcal{B}_{ij}(t)} (2\eta_b^{\text{A}} - 1)^2 / |\mathcal{B}_{ij}(t)|$.*

According to Theorem 1, we can determine the order of the round-average regret and its sublinearity.

**Corollary 1** (Sublinear Regret). *The round-average regret of Algorithm 1 is sublinear with an order of $\mathcal{O}(\sqrt{B \log(T)/T}/\Gamma)$.*

This sublinearity result is consistent with and generalizes those existing works on dueling bandits without considering evaluators' bias (e.g., [13]). Importantly, $\Gamma$ reflects the average deviation of the evaluators from spammers. When $\Gamma$ is larger (i.e., evaluators tend to reveal feedback either matching or opposing the ground-truth), the round-average regret is smaller.

## 4 Unknown Bias Case

We now solve the case where evaluators' bias is unknown to the agent, and the bias of any evaluator $\eta_m$ belongs to an infinite set $[0, 1]$. Our approach can be extended to the scenario with finite set of bias by projecting the continuous estimated bias to discrete space. Since the set of evaluators is finite, their bias comprises a finite set $\mathcal{B} \triangleq \{\eta_1, \eta_2, ..., \eta_M\}$. Let $N_{ij}^m(t)$ denote the number of feedback sent by evaluator $m$ and claiming $o_i \succ_m o_j$. Let $\mathcal{M}_{ij}(t) \subseteq \mathcal{M}$, which can be interpreted as the set of evaluators $m$ such that $N_{ij}^m(t) + N_{ji}^m(t) > 0$ for each pair of arms $i$ and $j$. Let $\mathcal{J}_m(t)$ denote the set of $(i, j)$ pairs such that $N_{ij}^m(t) + N_{ji}^m(t) > 0$ for each $m \in \mathcal{M}$.

Designing the extended bias-sensitive algorithm is highly non-trivial. This is because the estimation of the arm performance and evaluation bias is highly coupled. In the following, we first present the estimators for arm performance and evaluators' bias. Then, we propose the extended bias-sensitive algorithm that overcomes the aforementioned challenges. Finally, we analyze its regret.

## 4.1 Arm Performance and Bias Estimation

Let $\hat{p}_{ij}(t)$ and $\hat{\eta}_m(t)$ denote the estimation of $p_{ij}$ and $\eta_m$ given our estimator, respectively. After time slot $t$, the pairwise winning probability $p_{ij}$ in (1) is estimated using the same estimator as in Lemma 1 while replacing the ground-truth $\eta_m$ with the estimated $\hat{\eta}_m(t)$, i.e.,

$$\hat{p}_{ij}(t) = \frac{\sum_{m \in \mathcal{M}_{ij}(t)} (\bar{p}_{ij}^m(t) - (1 - \hat{\eta}_m(t))) (2\hat{\eta}_m(t) - 1)}{\sum_{m \in \mathcal{M}_{ij}(t)} (2\hat{\eta}_m(t) - 1)^2}. \tag{16}$$

Based on a similar idea as estimating the arm performance in Section 3.1, we formulate the problem for estimating the bias of evaluator $m \in \mathcal{M}$:

$$\hat{\eta}_m(t) = \arg\min_\eta \frac{1}{2} \|U_m \eta + \mathbf{b}^m\|^2 + \frac{\gamma}{2} \|\eta - \bar{\eta}_m\|^2. \tag{17}$$

In the first term, $U_m = (2\hat{p}_{ij}(t) - 1, i, j \in \mathcal{J}_m(t))$, and $\mathbf{b}^m = (1 - \hat{p}_{ij}(t) - \bar{p}_{ij}^m(t), i, j \in \mathcal{J}_m(t))$, where $\bar{p}_{ij}^m(t) \triangleq N_{ij}^m(t)/(N_{ij}^m(t) + N_{ji}^m(t))$. It aims to find the best $\eta$ that minimizes the estimation error of bias given the recent $\hat{p}_{ij}(t)$, similar as that in (9). The second term is introduced for the algorithm to be proposed. Its goal is to restrict the gap between the previous estimation $\bar{\eta}_m$ and the new estimation, where $\gamma$ balances the two terms. Solving (17) via the KKT conditions yields the estimator.

**Lemma 2** (Bias Estimator). *After time slot $t$, the bias of evaluator $\eta_m$ in (3) is estimated by*

$$\hat{\eta}_m(t) = \frac{\sum_{i,j \in \mathcal{J}_m(t)} (2\hat{p}_{ij}(t) - 1)(\bar{p}_{ij}^m(t) + \hat{p}_{ij}(t) - 1) + \gamma \bar{\eta}_m}{\sum_{i,j \in \mathcal{J}_m(t)} (2\hat{p}_{ij}(t) - 1)^2 + \gamma}. \tag{18}$$

Estimators (16) and (18) form a system of equations, and solving them jointly yields $\hat{p}_{ij}(t)$ and $\hat{\eta}_m(t)$. However, $\hat{p}_{ij}(t)$ and $\hat{\eta}_m(t)$ are highly coupled, i.e., the performance estimates depend on the bias estimates and vice versa, and the joint estimation problem is non-convex. Although it is possible to let $\hat{p}_{ij}(t)$ and $\hat{\eta}_m(t)$ update iteratively using (16) and (18), parameter $\hat{p}_{ij}(t)$ usually converges to local optimal solution $\hat{p}_{ij}(t) = 0.5$ due to the non-convexity. To address this, we decouple the evaluation bias from the arm performance estimation when initializing the estimation in each time slot and propose a BCD-based algorithm [23].

## 4.2 Extended Bias-Sensitive UCB Algorithm

We present the extended bias-sensitive UCB algorithm. Its pseudocode is given in Algorithm 2 of Appendix G. At the beginning of time slot $t$, estimators $\hat{p}_{ij}(t-1)$ and $\hat{\eta}_m(t-1)$ are computed. Specifically, estimator $\hat{p}_{ij}(t-1)$ is first set to $\bar{p}_{ij}^m(t-1) \triangleq N_{ij}^m(t-1)/(N_{ij}^m(t-1) + N_{ji}^m(t-1))$, i.e., the estimation of arm performance ignoring the evaluators' bias. This process decouples the impact of evaluators' bias estimation and that of inaccurate performance and bias estimation in the past time slots. Based on this $\hat{p}_{ij}(t-1)$, estimator $\hat{\eta}_m(t-1)$ is computed using (18). Then, according to BCD [23], $\hat{p}_{ij}(t-1)$ and $\hat{\eta}_m(t-1)$ are updated in sequence twice. We empirically show in Appendix K.7 that performing such updates twice leads to the best performance.[5] Either increasing or decreasing the rounds of updates leads to regret increase.

After that, the agent estimates the confidence radius $\hat{r}_{ij}(t-1)$ with the estimated bias $\hat{\eta}_m(t-1)$:

$$\hat{r}_{ij}(t-1) = \frac{\sum_{m \in \mathcal{M}_{ij}(t)} |2\hat{\eta}_m(t-1) - 1| \sqrt{\frac{\alpha \log(t-1)}{(N_{ij}^m(t-1) + N_{ji}^m(t-1))}}}{\sum_{m \in \mathcal{M}_{ij}(t-1)} (2\hat{\eta}_m(t-1) - 1)^2}. \tag{19}$$

Note that this is not the actual confidence radius for the estimators and thus leads to additional regret in decision making (see Section 4.3). Finally, $\hat{p}_{ij}(t-1)$ and $\hat{r}_{ij}(t-1)$ are substituted into (13) to compute $\text{UCB}_i(t)$, and the arms that optimize problem (14) are selected.

---

[5]Note that these steps on the initialization and first update of $\hat{p}_{ij}(t-1)$ and $\hat{\eta}_m(t-1)$ in each time slot are used to stabilize the estimation and can be skipped after a certain number of time slots once the estimation is relatively accurate for convergence acceleration. We empirically find that such a time slot threshold can be set in the form of $cK \log K$, where $c$ is a tunable coefficient.

## 4.3 Regret Analysis

We first quantify the actual confidence radius under estimators in (16) and (18), with which we are able to bound the regret of Algorithm 2. The proof is given in Appendix H.

**Lemma 3** (Confidence Radius). *Given the estimators in* (16) *and* (18)*, the confidence radius is*

$$r_{ij}^{\circ}(t) = \frac{\sum_{m\in\mathcal{M}_{ij}(t)}\left|\phi_{ij}^m(t)/(\epsilon_m^\eta(t))^2 - \hat{\phi}_{ij}^m(t)\right|}{\sum_{m\in\mathcal{M}_{ij}(t)}(2\hat{\eta}_m(t)-1)^2} + \frac{\sum_{m\in\mathcal{M}_{ij}(t)}\frac{|2\hat{\eta}_m(t)-1|}{|\epsilon_m^\eta(t)|}\sqrt{\frac{\alpha\log(t)}{N_{ij}^m(t)+N_{ji}^m(t)}}}{\sum_{m\in\mathcal{M}_{ij}(t)}(2\hat{\eta}_m(t)-1)^2}, \quad (20)$$

*where* $\phi_{ij}^m(t) \triangleq (\bar{p}_{ij}^m(t) - (1-\eta_m))/(2\eta_m - 1)$, $\hat{\phi}_{ij}^m(t) \triangleq (\bar{p}_{ij}^m(t) - (1-\hat{\eta}_m(t)))/(2\hat{\eta}_m(t)-1)$, $\epsilon_m^\eta(t) \triangleq (2\eta_m-1)/(2\hat{\eta}_m(t)-1)$.

Quantifying the regret using the difference between $\hat{r}_{ij}(t)$ and $r_{ij}^{\circ}(t)$ is challenging, because the mapping from the confidence radius to the exact probability an estimation falls within the radius can hardly be solved, due to the complex form of estimators. Thus, we define a parameter $\xi(t)$.

**Definition 3** (Parameter $\xi(t)$). *For each time slot $t$, let $\xi(t)$ denote the minimum non-negative value such that* $\xi(t) \geq \frac{1}{2}\left(\mathrm{HF}\left(\mathbb{P}\left(|\hat{p}_{ij}(t) - p_{ij}| \leq r_{ij}^{\circ}(t)\right)\right) - \mathbb{P}\left(|\hat{p}_{ij}(t) - p_{ij}| \leq \hat{r}_{ij}(t)\right)\right)$, *where* $\mathrm{HF}(\cdot)$ *is the tight lower bound of* $\mathbb{P}\left(|\hat{p}_{ij}(t) - p_{ij}| \leq r_{ij}^{\circ}(t)\right)$.

As will be seen in Theorem 2, a lower $\xi(t)$ leads to a lower regret. There are various cases that ensure $\xi(t) = 0$. Although it is hard to derive all the cases due to the complex form of $\hat{r}_{ij}(t)$ and $r_{ij}^{\circ}(t)$, we list two examples: (i) $\hat{r}_{ij}(t) = r_{ij}^{\circ}(t)$; (ii) $\eta_m(t) = 1$ and $\hat{\eta}_m(t) \in (0.5, 1]$ for all $m \in \mathcal{M}$.

Then, the round-average regret can be determined, with the proof given in Appendix I.

**Theorem 2** (Regret of Extended Bias-Sensitive Algorithm). *Under Definition 3, the extended bias-sensitive UCB algorithm based on estimators in* (16) *and* (18) *has a round-average regret of*

$$\frac{1}{T}\sum_{t=1}^{T}\boldsymbol{Reg}(\boldsymbol{x}(t)) \leq \frac{\overline{UCB}\left(K(K-1) + 2\left(H + \sum_{t=K(K-1)/2+1}^{T}\xi(t)\right)\right)}{T}$$

$$+ \frac{2\overline{UCB}\sqrt{\alpha\log(T)}}{\Gamma}\left(\frac{H + B^2\log(BT^{2\alpha})}{T} + \sqrt{\frac{2BK}{K-1}}\cdot\frac{1}{\sqrt{T}}\right). \quad (21)$$

When compared with Theorem 1 for known bias case, the round-average regret under unknown bias case has the same order but incorporates an additional bounded error related to $\xi(t)$. If $\xi(t)$ is monotonically decreasing and converges to zero as $t \to \infty$, then this bounded error approaches zero. However, due to the non-convexity of the joint estimation problem, proving this convergence is an open problem under BCD. In Appendix J, we empirically show that this bounded error is small and can approach zero.

## 5 Experiments

We consider that the user bias follows a Beta distribution $\mathrm{Beta}(\alpha_B, \beta_B)$ [21]. We use the BT model to model the winning probability of arms [21], i.e., $p_{ij} = e^{s_i}/(e^{s_i} + e^{s_j})$, where $s_i$ is a coefficient associated with arm $i$ following Gaussian distribution $\mathcal{N}(\mu, \sigma^2)$ and a Condorcet winner typically exists under this model. Unless otherwise specified, we set $\eta_m \sim \mathrm{Beta}(\alpha_B = 2, \beta_B = 1)$ and $s_i \sim \mathcal{N}(\mu = 0, \sigma^2 = 2)$. Through empirical tests, we set $\alpha = \alpha_0(\sum_{m\in\mathcal{M}_{ij}(t)}(2\eta_m - 1)^2)^2/(\sum_{m\in\mathcal{M}_{ij}(t)}|2\eta_m - 1|)^2$, where $\alpha_0 = 0.51$ [11] and $\eta_m$ can be the recent estimated value for unknown bias case. The term $\alpha$ relies on the recent estimation of $\eta_m$ and helps to mitigate the over-exploration due to the presence of evaluators' bias. We set coefficient $c = 50$. Our code is built based on open source code [28] for dueling bandits. Experiments are conducted on a compute platform with an AMD Ryzen 7 7800X3D (8-core) processor and 64 GB of RAM (4800 MHz). We run each experiment for 100 times and show the average results in this section. The results with standard error can be found in Appendix K.

We compare our algorithms with five baselines in dueling bandits: Relative Confidence (denoted by "RC") [10], Relative UCB (denoted by "RUCB") [11], a Bayesian method Double Thompson

| | Cumulative Average Regret (↓) | | | | | | Cumulative Weak Regret (↓) | | | | | |
|---|---|---|---|---|---|---|---|---|---|---|---|---|
| | Arm Heter. $\sigma^2$ | | | Bias Concentr. $\alpha_{\mathrm{B}}$ | | | Arm Heter. $\sigma^2$ | | | Bias Concentr. $\alpha_{\mathrm{B}}$ | | |
| | 1.0 | 2.0 | 4.0 | 1.0 | 2.0 | 3.0 | 1.0 | 2.0 | 4.0 | 1.0 | 2.0 | 3.0 |
| RC | 1374 | 1338 | 967 | 2845 | 1338 | 687 | 596 | 525 | 502 | 1847 | 525 | 278 |
| RUCB | 1906 | 2134 | 1154 | 2832 | 2134 | 1185 | 1018 | 1144 | 719 | 1829 | 1144 | 506 |
| DT | 1396 | 1425 | 942 | 2621 | 1425 | 640 | 445 | 492 | 375 | 1428 | 492 | 191 |
| MBTW | 1220 | 1509 | 726 | 1769 | 1509 | 1448 | 175 | 162 | 140 | 569 | 162 | 92 |
| UCB | 1283 | 1426 | 732 | 2581 | 1426 | 706 | 553 | 548 | 336 | 1583 | 548 | 153 |
| **RC-B**(*) | 1378 | 1611 | 1050 | 2207 | 1611 | 803 | 649 | 869 | 727 | 1119 | 869 | 502 |
| **RUCB-B**(*) | 993 | 1120 | 709 | 1191 | 1120 | 1055 | 422 | 480 | 370 | 604 | 480 | 446 |
| **DT-B**(*) | **430** | **411** | **344** | 631 | **411** | **280** | 198 | 210 | 168 | 436 | 210 | 110 |
| **BS-UN**(*) | 690 | 689 | 387 | 825 | 689 | 637 | 194 | 161 | 94 | 340 | 161 | 92 |
| **BS-K**(*) | 654 | 713 | 407 | **554** | 713 | 624 | **116** | **90** | **79** | **60** | **90** | **82** |

Table 1: Performance under diverse arm heterogeneity (denoted by "heter.") and bias concentration (denoted by "concentr.") with 10 arms and 10 evaluators. **Our methods are marked with "(*)".** The best, second, and third best results are marked in bold, underline, and dashed underline, respectively.

(denoted by "DT") [12], Modified Beat The Winner (denoted by "MBTW") [27], UCB (which follows [13] but omits the cost constraint). Meanwhile, we incorporate our bias-sensitive estimation in Algorithm 2 and obtain bias-sensitive versions of RC, RUCB, and DT (see Appendix K.1). They are denoted by "[Baseline Name]-B". Our Algorithms 1 and 2 are denoted by BS-K and BS-UN for known and unknown bias cases, respectively. We use "(*)" to mark our methods (including the bias-sensitive versions of baselines and our proposed BS-K and BS-UN). Experiments are conducted under unknown bias case, expect for those of BS-K. We show the cumulative regret $\sum_{t=1}^{T} \mathbf{Reg}(\boldsymbol{x}(t))$ because (i) the values of round-average regret are very small, and (ii) cumulative regret can infer marginal regret in figures. In Table 1, the round-average regret can be obtained by dividing the cumulative regret by $T = 10000$.

**Algorithm Comparison:** Tables 1 and 2 show the cumulative regret after 10000 rounds. The algorithm convergence and standard error are shown in Appendix K.2. The numerical bias estimation error can be found in Appendix K.3. We have the following observations. (i) Our proposed BS-UN and BS-K algorithms achieve superior performance under both average and weak regrets, ranked top three among all algorithms for most cases. When compared with RC, RUCB, DT, MBTW, and UCB, the average regret reduction of BS-UN can be up to 71.0%, 70.9%, 68.5%, 56.0%, and 68.0%, respectively; the weak regret reduction of BS-UN can be up to 81.6%, 86.9%, 76.2%, 40.2%, 78.5%, respectively. (ii) The bias-sensitive versions of baselines usually achieve lower regret than their original versions, showing the effectiveness of our estimators. For RUCB and DT, their average regret reduction can be up to 58.0% and 75.9%, respectively; their weak regret reduction can be up to 67.0% and 69.5%, respectively.

**Impact of $\alpha_{\mathrm{B}}$:** Our BS-K and BS-UN algorithms are more beneficial when bias concentration $\alpha_{\mathrm{B}}$ is lower. Specifically, a smaller $\alpha_{\mathrm{B}}$ implies a higher degree of evaluators' bias. In Table 1, when $\alpha_{\mathrm{B}}$ reduces from 3.0 to 2.0, the average and weak regrets of baselines increase by up to $1.23$ times and $2.58$ times, respectively. However, the average and weak regret increasing are $0.08$ and $0.75$ for BS-UN and $0.14$ and $0.09$ for BS-K, respectively.

**Impact of $\sigma^2$:** A larger $\sigma^2$ implies a higher degree of arm heterogeneity. Since the evaluators' bias is not accounted by the baselines, a larger heterogeneity makes it easier to identify the best arm and hence a lower regret. As the evaluators' bias is accounted by our methods, a moderate heterogeneity can be sufficient for identifying the best arm and reducing the regrets.

**Impact of Evaluators and Arms:** From Table 2, (i) our methods are not sensitive to the number of evaluators. As the number of evaluators increases from 5 to 20, the average and weak regrets of our BS-UN increase by $-0.10$ and $-0.35$ times, respectively; those of our BS-K increase by $0.08$ and $0.32$ times, respectively. (ii) The increasing in the number of arms increases the regrets of our methods. This is acceptable, because in the LLM evaluation example, the number of users (i.e., evaluators) is always large, while the number of LLMs (i.e., arms) is usually small, e.g., around 10. We further evaluate large-scale settings with 100 evaluators and 100 arms in Appendix K.4.

| Method | Cum. Average Regret (↓) | | | | | | Cum. Weak Regret (↓) | | | | | |
|---|---|---|---|---|---|---|---|---|---|---|---|---|
| | Num. of Eval. | | | Num. of Arms | | | Num. of Eval. | | | Num. of Arms | | |
| | 5 | 15 | 20 | 5 | 15 | 20 | 5 | 15 | 20 | 5 | 15 | 20 |
| RC | 1590 | 1142 | 1301 | 561 | 2250 | 2568 | 689 | 409 | 497 | 85 | 1303 | 1626 |
| RUCB | 2293 | 1924 | 2042 | 813 | 2277 | 2499 | 1272 | 979 | 1067 | 181 | 1356 | 1593 |
| DT | 1548 | 1073 | 1221 | 695 | 1854 | 2006 | 491 | 295 | 349 | 142 | 702 | 802 |
| MBTW | 1444 | 1452 | 1509 | 1218 | 1260 | 1330 | 177 | 124 | 182 | 48 | 307 | 438 |
| UCB | 1604 | 1099 | 1252 | 631 | 1752 | 2115 | 709 | 387 | 491 | 77 | 971 | 1264 |
| **RC-B**(*) | 1801 | 1370 | 1694 | 575 | 2301 | 2707 | 934 | 785 | 1005 | 198 | 1492 | 1993 |
| **RUCB-B**(*) | 964 | 1102 | 1115 | 820 | 1360 | 1868 | 392 | 468 | 483 | 243 | 707 | 1146 |
| **DT-B**(*) | **360** | **375** | **344** | **169** | **736** | 1156 | 177 | 165 | 169 | 77 | 308 | 578 |
| **BS-UN**(*) | 688 | 722 | 621 | 626 | 929 | 1157 | 121 | 97 | 79 | 108 | 364 | 543 |
| **BS-K**(*) | 588 | 600 | 638 | 469 | 881 | **1021** | 57 | 73 | 75 | 21 | 306 | 420 |

Table 2: Impact of the number of evaluators (denoted by "Eval.") and arms. Unless specified in the column title, the default number of arms and evaluators is 10. **Our methods are marked with "(*)".**

**Ablation Study:** When compared with the alternative estimator given in Appendix A, our arm performance estimators in (10) and (16) reduce the average regret by $58.3\%$ and $42.1\%$ and weak regret by $78.1\%$ and $61.6\%$ for known and unknown bias cases, respectively. When compared with other estimators (e.g., estimators based on conditional probability expression and other estimator update procedures), our bias estimators in (18) reduces the average and weak regrets by $11.2\% - 49.1\%$ and $13.1\% - 75.5\%$, respectively. Please refer to Appendix K.6 for details.

## 6 Conclusion and Limitations

This work presents the first study on addressing evaluators' bias in dueling bandits. We overcome the challenge of non-convexity and bias heterogeneity and propose bias-sensitive algorithms with regret bounds. When compared with baselines, our algorithms reduces the regret by up to $86.9\%$, especially when the evaluators' bias levels are more heterogeneous. Meanwhile, our proposed estimator can be incorporated into baselines and achieve a regret reduction of up to $75.9\%$.

The main limitations of this work contain four parts. First, the bias of each evaluator is modeled to be deterministic and unchanged across time. To extend the model to stochastic and diverse bias, we may learn from adversarial dueling bandits and extend the techniques from addressing time-varying winning probability to time-varying bias. Second, the regret bound for BS-UN contains a term $\xi(t)$, which was not derived in closed form. It is important to derive the specific expression of it to reveal further insights, overcoming the difficulty in analyzing the performance of a BCD algorithm for non-convex problem. Third, the recent bias modeling is only evaluator-dependent. It is interesting to consider arm-dependent bias, characterizing evaluators' distinctive bias toward arms. Fourth, this work is motivated by human bias in feedback. It would be beneficial to construct real-world experiments with humans for algorithm evaluation.

## Acknowledgments and Disclosure of Funding

This work was supported in part by the National Natural Science Foundation of China under Grant 62202214 and Guangdong Basic and Applied Basic Research Foundation under Grant 2023A1515012819.

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

## Appendix A    An Alternative Estimator

It is possible to derive an alternative unbiased estimator by directly substituting $\bar{p}^b_{ij}(t) \triangleq N^b_{ij}(t)/N^b_{ij}(t) + N^b_{ji}(t)$ as $p^m_{ij}$ for $\eta_m = \eta^{\text{A}}_b$ into (4). This leads to the following estimator:

**Definition 4** (An Alternative Estimator). *After time slot t, we propose to approximate $p_{ij}$ in (1) using*

$$\hat{p}_{ij}(t) = \frac{1}{|\mathcal{B}_{ij}(t)|} \sum_{b \in \mathcal{B}_{ij}(t)} \frac{\frac{N^b_{ij}(t)}{N^b_{ij}(t)+N^b_{ji}(t)} - (1 - \eta^{\text{A}}_b)}{2\eta^{\text{A}}_b - 1}. \tag{22}$$

Although we can prove that this alternative estimator is unbiased, this estimator $\hat{p}_{ij}(t)$ approaches infinite when there exist evaluators that are spammers, i.e., when there exists $b \in \mathcal{B}$ such that $\eta^{\text{A}}_b = 0.5$. This is not practical, as we cannot prevent the existence of spammers in practical systems.

Formally, under this estimator in Definition 4, the confidence radius and the associated round-average regret (i.e., the regret under our proposed bias-sensitive algorithm while replacing the estimator and confidence radius accordingly) can be determined as follows. The proofs are similar as those in the main context and hence omitted here.

**Lemma 4** (Confidence Radius under the Alternative Estimator). *The confidence radius $r_{ij}(t)$ in Definition 2 is determined by*

$$r_{ij}(t) = \frac{1}{|\mathcal{B}_{ij}(t)|} \sum_{b \in \mathcal{B}_{ij}(t-1)} \sqrt{\frac{\alpha \log(t)}{(N^b_{ij}(t) + N^b_{ji}(t))|2\eta^{\text{A}}_b - 1|}}. \tag{23}$$

**Lemma 5** (Regret under the Alternative Estimator). *The bias-sensitive UCB algorithm under the estimator in Definition 4 and confidence radius in Lemma 4 has a round-average regret of*

$$\frac{1}{T} \sum_{t=1}^{T} \boldsymbol{Reg}(\boldsymbol{x}(t)) \le \frac{K-1}{T} \left( \frac{(K-1)K}{2}(1 + \overline{UCB}) + H(1 + K\overline{UCB}) \right)$$

$$+ \frac{(K-1)K\overline{UCB}\sqrt{\alpha \log(T)}}{\widetilde{\Gamma}} \left( \left(1 - \frac{1}{|\mathcal{B}|}\right) + \frac{H_{|B|-1}}{T} + \left( \frac{2\sqrt{T-|\mathcal{B}|+1}}{T|\mathcal{B}|} - \frac{1}{T|\mathcal{B}|} \right) \right), \tag{24}$$

*where $\widetilde{\Gamma} \triangleq \min_{b \in \mathcal{B}} \sqrt{|2\eta^{\text{A}}_b - 1|}$.*

When comparing with the round-average regret of our proposed bias-sensitive algorithm in Theorem 1, this regret is different in terms of $\widetilde{\Gamma}$. As we can expect, if there exists any evaluator that is a spammer (i.e., $\eta^{\text{A}}_b = 0.5$), then $\widetilde{\Gamma}$ is equal to zero, making the round-average regret approaches infinite.

## Appendix B    Proof for Lemma 1

By substitute (10) into $\mathbb{E}[\hat{p}_{ij}(t)]$, we have

$$\mathbb{E}[\hat{p}_{ij}(t)] = \frac{\sum_{b \in \mathcal{B}_{ij}(t)} (2\eta^{\text{A}}_b - 1) \left( \mathbb{E}\left[ \frac{N^b_{ij}(t)}{N^b_{ij}(t)+N^b_{ji}(t)} \right] - (1 - \eta^{\text{A}}_b) \right)}{\sum_{b \in \mathcal{B}_{ij}(t)} (2\eta^{\text{A}}_b - 1)^2}. \tag{25}$$

Since event $o_i \succ_m o_j$ for any $t' \le t$ is a Bernoulli trial which holds with probability $\Pr(o_i \succ_m o_j)$, $\mathbb{E}[N^b_{ij}(t)/(N^b_{ij}(t) + N^b_{ji}(t))] = \Pr(o_i \succ_m o_j)$ for any $m$ ensuring $\eta_m = \eta^{\text{A}}_b$. By substituting (4) into (25), we have $\mathbb{E}[\hat{p}_{ij}(t)] = p_{ij}$.

## Appendix C    Proof for Proposition 1

To alleviate the coupling of the randomness among time slots, we introduce a $B \times t$ table. Cell $(b \in \mathcal{B}, s \in \mathcal{T})$ corresponds to the $s$-th times that $\{i, j\}$ is selected for evaluators of bias $b$. Let

$N_{ij}^{b,s}$ denote the number of time slots $t'$ (i) until (i.e., on and before) the $s$-th times that $\{i, j\}$ is selected for the evaluators with bias $\eta_b^{\text{A}}$ such that (ii) evaluator $m$ sends a feedback with $o_i \succ_m o_j$, and (iii) $\eta_{m_{t'}} = \eta_b^{\text{A}}$. Let $\overline{q}_{i,j}^{b,s}$ denote the fraction of feedback sent by the evaluators of bias $b$ claiming $o_i \succ_m o_j$ among the first $s$ times that $\{i, j\}$ is selected for the evaluators of bias $b$. That is,

$$\overline{q}_{ij}^{b,s} = \frac{N_{ij}^{b,s}}{s}. \tag{26}$$

Then, according to Hoeffding Inequality, for any $t > 0$,

$$\Pr\left(\left|\overline{q}_{ij}^{b,s} - \mathbb{E}[\overline{q}_{ij}^{b,s}]\right| \leq \sqrt{\frac{\alpha \log(t)}{s}}\right) \geq 1 - \frac{2}{t^{2\alpha}}, \tag{27}$$

where $\mathbb{E}[\overline{q}_{ij}^{b,s}]$ denotes the expected value of $\overline{q}_{ij}^{b,s}$ considering the randomness of evaluator feedback. According to (4), $\mathbb{E}[\overline{q}_{ij}^{b,s}]$ can be determined by

$$\mathbb{E}[\overline{q}_{ij}^{b,s}] = \eta_b^{\text{A}} p_{ij} + (1 - \eta_b^{\text{A}})(1 - p_{ij}). \tag{28}$$

Substituting (28) into the inequality on the left-hand side of (27), we have

$$|\overline{q}_{ij}^{b,s} - (\eta_b^{\text{A}} p_{ij} + (1 - \eta_b^{\text{A}})(1 - p_{ij}))| \leq \sqrt{\frac{\alpha \log(t)}{s}}. \tag{29}$$

Multiplying both sides by $|2\eta_b^{\text{A}} - 1|$ and substituting (26), inequality (29) can be transformed into

$$\left|\left(\frac{N_{ij}^{b,s}}{s} - (1 - \eta_b^{\text{A}})\right)(2\eta_b^{\text{A}} - 1) - p_{ij}(2\eta_b^{\text{A}} - 1)^2\right| \leq |2\eta_b^{\text{A}} - 1|\sqrt{\frac{\alpha \log(t)}{s}}, \tag{30}$$

Let $s = N_{ij}^b(t) + N_{ji}^b(t)$ and hence $N_{ij}^{b,s} = N_{ij}^b(t)$. Based on triangle inequality, considering (30) for all possible $b \in \mathcal{B}_{ij}(t)$, the following inequality holds:

$$\Pr\left(\left|\sum_{b \in \mathcal{B}_{ij}(t)} \left(\frac{N_{ij}^b(t)}{N_{ij}^b(t) + N_{ji}^b(t)} - (1 - \eta_b^{\text{A}})\right)(2\eta_b^{\text{A}} - 1) - \sum_{b \in \mathcal{B}_{ij}(t)} p_{ij}(2\eta_b^{\text{A}} - 1)^2\right|\right.$$

$$\left. \leq \sum_{b \in \mathcal{B}_{ij}(t)} |2\eta_b^{\text{A}} - 1|\sqrt{\frac{\alpha \log(t)}{N_{ij}^b(t) + N_{ji}^b(t)}}\right) \geq 1 - \frac{2}{t^{2\alpha}}. \tag{31}$$

Finally, dividing both sides of the inequality in $\Pr(\cdot)$ by $\sum_{b \in \mathcal{B}_{ij}(t)} |2\eta_b^{\text{A}} - 1|^2$, we show that $r_{ij}(t)$ defined in (11) is the confidence radius given the definition of $\hat{p}_{ij}(t)$.

## Appendix D  Bias-Sensitive UCB Algorithm

Algorithm 1 shows the pseudocode of our proposed bias-sensitive UCB Algorithm.

---
**Algorithm 1** Bias-Sensitive UCB Algorithm

---
1: **for** each time slot $t = 1$ to $T$ **do**
2:     Update $\hat{p}_{ij}(t-1)$ using (10) and $r_{ij}(t-1)$ using (11);
3:     Estimate $\text{UCB}_i(t)$ using (13) for $i \in \mathcal{K}$;
4:     Select the arms $x_1(t)$ and $x_2(t)$ that optimize problem (14);
5: **end for**

---

## Appendix E  An Alternative Algorithm that Selects the "Best Competitor"

In our algorithm, in each time slot $t$, the agent selects the arms with the highest and second highest UCB values as the first arm (denoted by $x_1(t)$) and the second arm (denoted by $x_2(t)$), respectively, i.e., $x_1(t) = \arg\max_{i \in \mathcal{K}} \text{UCB}_i(t)$ and $x_2(t) = \arg\max_{i \in \mathcal{K} \setminus \{x_1(t)\}} \text{UCB}_i(t)$. As an alternative,

Table 3: Methods considered for comparing the selection of the second arm.

| | First Arm | Second Arm | Arm/Bias Estimation |
|---|---|---|---|
| RUCB-B | Randomly pick from an optimistic arm pool | "Strongest competitor" | RUCB-B/Our |
| BS-K-Modified | Highest UCB | "Strongest competitor" | Our/Known Bias |
| BS-K | Highest UCB | Second highest UCB | Our/Known Bias |
| BS-UN-Modified | Highest UCB | "Strongest competitor" | Our/Our |
| BS-UN | Highest UCB | Second highest UCB | Our/Our |

Table 4: Comparison of the selection of the second arm under BT Model. Different columns correspond to different number of arms.

| | Cumulative Average Regret | | | | Cumulative Weak Regret | | | |
|---|---|---|---|---|---|---|---|---|
| | 10 | 20 | 30 | 50 | 10 | 20 | 30 | 50 |
| RUCB-B | 1943 | 2914 | 4280 | 8990 | 685 | 1444 | 2451 | 6468 |
| BS-Modified-K | **756** | **1308** | **1592** | 5155 | 59 | **322** | 644 | 1732 |
| BS-K | 868 | 1361 | 1748 | **4440** | **57** | 336 | **537** | **1256** |
| BS-Modified-UN | **829** | 1469 | 1951 | 5643 | 96 | **418** | 815 | 2291 |
| BS-UN | 990 | **1461** | **1834** | **4109** | **92** | 477 | **687** | **1355** |

the agent may choose the "strongest competitor" of the first arm as the second arm, i.e., $x_2(t) = \arg\max_{i \in \mathcal{K} \setminus \{x_1(t)\}} \text{UCB}_{ix_1(t)}(t)$. In the following, we empirically show that for the second arm, considering the second best arm (i.e., the one with the second highest UCB) and the "strongest competitor" of the first arm achieve similar performance for many of the cases, while the former can achieve better performance when the number of arms is large or a Condorcet winner does not exist.

To conduct such experiments, we compare five methods as shown in Table 3. Specifically, RUCB-B is built upon Relative UCB [11] while incorporating our bias estimation method. BS-K-Modified and BS-UN-Modified correspond to the methods choosing the "strongest competitor" of the first arm as the second arm, incorporated with our arm and bias estimation methods. BS-K and BS-UN are our proposed approaches.

**Bradley-Terry (BT) Model:** Table 4 shows the results under the same arm performance modeling as the main experimental results, i.e., where BT model is considered. In this case, a Condorcet winner always exists. First, for many of the cases, BS-K-Modified and BS-K (as well as BS-UN-Modified and BS-UN) achieve similar average and weak regrets. This indicates that those two methods for choosing the second arm do not make significance difference. Second, when there are 50 arms, our BS-K and BS-UN always outperform BS-K-Modified and BS-UN-Modified, respectively. This implies that independently selecting two arms (rather than having the selection of the second arm rely on the first arm) for exploration is more beneficial for arm performance estimation when the number of arms is large.

**Non-Existence of a Condorcet Winner:** The arm performance matrix is initialized with the BT model. To remove the Condorcet winner, for each arm, we randomly select two arms that are initially weaker than this arm and increase their winning probabilities (that beat this arm) to a random value within range $(0.5, 0.6)$. In Table 5, when a Condorect winner does not exist, our BS-K and BS-UN always outperform BS-K-Modified and BS-UN-Modified in terms of the average regret, respectively. In this case, the "strongest competitor" of the first arm may perform badly when compared with other arms, so selecting the "strongest competitor" as the second arm may lead to a high regret and hence increase the average rerget. RUCB-B achieves the worst performance. This result shows that the choice of the first arm makes more significant impact than that of the second arm.

## Appendix F    Proof for Theorem 1

We first present the proof details. Then, we prove an auxiliary inequality used in the proof.

Table 5: Comparison of the selection of the second arm under non-existence of a Condorcet winner. Different columns correspond to different number of arms.

| | Cumulative Average Regret | | | | Cumulative Weak Regret | | | |
|---|---|---|---|---|---|---|---|---|
| | 10 | 20 | 30 | 50 | 10 | 20 | 30 | 50 |
| RUCB-B | 3619 | 3966 | 4918 | 8619 | 1133 | 1913 | 2841 | 6238 |
| BS-Modified-K | 2602 | 2387 | 2625 | 5205 | 224 | **476** | 865 | 2297 |
| BS-K | **1715** | **1787** | **2146** | **4737** | **198** | 588 | **755** | **1631** |
| BS-Modified-UN | 2742 | 2701 | 2808 | 5418 | **276** | 811 | 990 | 2661 |
| BS-UN | **1401** | **1768** | **2186** | **4648** | 437 | **697** | **907** | **1960** |

### F.1 Proof Details

This proof essentially derives the bound for average regret, while it works for weak regret by relaxing the weak regret to average regret in inequality (a) of (33).

Based on the definition of average regret $\mathbf{RegA}(\boldsymbol{x}(t))$,

$$\sum_{t=1}^{T} \mathbf{RegA}(\boldsymbol{x}(t)) = \sum_{t=1}^{T} \mathbb{E}\left[\theta_{i^*} - (\theta_{x_1(t)} + \theta_{x_2(t)})/2\right]. \tag{32}$$

We define $\mathcal{I}(t) \triangleq \{x_1(t), x_2(t)\}$ as the set of arms that are selected in time slot $t$. We determine the upper bound as follows:

$$\begin{aligned}
&\theta_{i^*} - (\theta_{x_1(t)} + \theta_{x_2(t)})/2 \\
&\overset{(a)}{\leq} \theta_{i^*} - \tfrac{1}{2}\sum_{i\in\mathcal{I}(t)} \theta_i + \tfrac{1}{2}\sum_{i\in\mathcal{I}(t)} \left(\mathrm{UCB}_i(t) - \mathrm{UCB}_{i^*}(t)\right) \\
&\overset{(b)}{\leq} \tfrac{1}{2}\sum_{i\in\mathcal{I}(t)} \left(\mathrm{UCB}_i(t) - \theta_i\right) + \theta_{i^*} - \mathrm{UCB}_{i^*}(t) \\
&\overset{(c)}{\leq} \tfrac{1}{K-1}\sum_{i\in\mathcal{I}(t)}\sum_{j\neq i} \frac{\mathbf{UCB}_{ij}(t) - p_{ij}}{2} \\
&\quad + \tfrac{2}{K-1}\sum_{j\neq i^*} \frac{p_{i^*j} - \mathrm{UCB}_{i^*j}(t)}{2}.
\end{aligned} \tag{33}$$

Inequality (a) holds because arm $i \in \mathcal{I}(t)$ is selected by the algorithm and hence $\mathrm{UCB}_i(t) - \mathrm{UCB}_{i^*}(t) \geq 0$. Note that if weak regret is considered, we can relax the weak regret to average regret using $\max\{\theta_{x_1(t)}, \theta_{x_2(t)}\} \geq \sum_{i\in\mathcal{I}(t)} \theta_i/2$ in (a). Inequality (b) holds by rearranging the terms. Inequality (c) holds based on the definition of $\theta_i$ in (2) and $\mathrm{UCB}_i(t)$ in (13). Let $\Phi_1(t) \triangleq \sum_{i\in\mathcal{I}(t)}\sum_{j\neq i} (\mathbf{UCB}_{ij}(t) - p_{ij})/2$ and let $\Phi_2(t) \triangleq \sum_{j\neq i^*} (p_{i^*j} - \mathrm{UCB}_{i^*j}(t))/2$. Thus,

$$\sum_{t=1}^{T} \mathbb{E}\left[\theta_{i^*} - (\theta_{x_1(t)} + \theta_{x_2(t)})/2\right] \leq \frac{1}{K-1}\sum_{t=1}^{T} \mathbb{E}[\Phi_1(t)] + \frac{2}{K-1}\sum_{t=1}^{T} \mathbb{E}[\Phi_2(t)]. \tag{34}$$

In the following, we will bound the above two terms $\sum_{t=1}^{T}\mathbb{E}[\Phi_1(t)]$ and $\sum_{t=1}^{T}\mathbb{E}[\Phi_2(t)]$, respectively.

**Bound $\sum_{t=1}^{T}\mathbb{E}[\Phi_1(t)]$:** At the beginning of Algorithm 1, each dueling pair will be selected once. This holds because at the beginning of the algorithm, the arms have not been explored such that that UCBs cannot been computed, so their UCBs' are initially set to be large values to enable the exploration. This is consistent with existing works [11, 13]. These selections of each dueling pair take a total of $t_0 = C(K, 2)$ rounds. Thus, the regret during rounds 1 to $t_0$ is given as follows:

$$\sum_{t=1}^{t_0} \mathbb{E}[\Phi_1(t)] \leq (K-1)\overline{\mathrm{UCB}}t_0, \tag{35}$$

where $\overline{\mathrm{UCB}}$ is the upper limit of $\mathrm{UCB}_{ij}(t)$ for $i, j \in \mathcal{N}$.

Then, we focus on the rounds after $t_0$, i.e., $t = \{t_0 + 1, ..., T\}$. During these rounds, we define the following events for models $i$ and $j$:

- $\mathcal{E}_{ij}(t)$: $\hat{p}_{ij}(t) - p_{ij} > r_{ij}(t)$;

- $\overline{\mathcal{E}}_{ij}(t)$: complement of $\mathcal{E}_{ij}(t)$.

In particular, event $\mathcal{E}_{ij}(t)$ corresponds to the case where $\hat{p}_{ij}(t)$ has been overestimated and exceeds the upper confidence bound. This case induces $\mathbf{UCB}_{ij}(t) \geq p_{ij}$. When event $\overline{\mathcal{E}}_{ij}(t)$ happens, $\mathbf{UCB}_{ij}(t) - p_{ij} \leq 2r_{ij}(t)$. Thus, we can bound $\sum_{t=t_0+1}^{T} \mathbb{E}[\Phi_1(t)]$ as follows:

$$\sum_{t=t_0+1}^{T} \mathbb{E}[\Phi_1(t)] \leq \sum_{t=t_0+1}^{T} \mathbb{E}\left[ \sum_{i \in \mathcal{I}(t)} \sum_{j \neq i} \frac{\mathbf{UCB}_{ij}(t) - p_{ij}}{2} \times \mathbf{1}(\mathbf{UCB}_{ij}(t) \geq p_{ij}) \right], \tag{36}$$

where $\mathbf{1}(\cdot)$ is the indicator function, i.e., $\mathbf{1}(x > y) = 0$ if $x > y$, and $\mathbf{1}(x > y) = 0$ otherwise. Since exactly one of events $\mathcal{E}_{ij}(t)$ and $\overline{\mathcal{E}}_{ij}(t)$ happens,

$$\mathbb{E}\left[ \sum_{i \in \mathcal{I}(t)} \sum_{j \neq i} \frac{\mathbf{UCB}_{ij}(t) - p_{ij}}{2} \mathbf{1}(\mathbf{UCB}_{ij}(t) \geq p_{ij}) \right]$$

$$= \mathbb{E}\left[ \underbrace{\sum_{i \in \mathcal{I}(t)} \sum_{j \neq i} \frac{\mathbf{UCB}_{ij}(t) - p_{ij}}{2} \mathbf{1}(\mathbf{UCB}_{ij}(t) \geq p_{ij}) \mathbf{1}(\mathcal{E}_{ij}(t))}_{\mathbb{E}[\Phi_{1,1}(t)]} \right]$$

$$+ \mathbb{E}\left[ \underbrace{\sum_{i \in \mathcal{I}(t)} \sum_{j \neq i} \frac{\mathbf{UCB}_{ij}(t) - p_{ij}}{2} \mathbf{1}(\mathbf{UCB}_{ij}(t) \geq p_{ij}) \mathbf{1}(\overline{\mathcal{E}}_{ij}(t))}_{\mathbb{E}[\Phi_{1,2}(t)]} \right].$$

$$\tag{37}$$

Substituting (37) into (36),

$$\sum_{t=t_0+1}^{T} \mathbb{E}[\Phi_1(t)] \leq \sum_{t=t_0+1}^{T} \mathbb{E}[\Phi_{1,1}(t)] + \sum_{t=t_0+1}^{T} \mathbb{E}[\Phi_{1,2}(t)]. \tag{38}$$

We now bound these two terms $\sum_{t=t_0+1}^{T} \mathbb{E}[\Phi_{1,1}(t)]$ and $\sum_{t=t_0+1}^{T} \mathbb{E}[\Phi_{1,2}(t)]$ respectively.

Bound $\sum_{t=t_0+1}^{T} \mathbb{E}[\Phi_{1,1}(t)]$: Since event $\mathcal{E}_{ij}(t)$ implies $\mathbf{UCB}_{ij}(t) \geq p_{ij}$, we have

$$
\begin{aligned}
& \textstyle\sum_{t=t_0+1}^{T} \mathbb{E}[\Phi_{1,1}(t)] \\
={} & \textstyle\sum_{t=t_0+1}^{T} \mathbb{E}\left[ \sum_{i \in \mathcal{I}(t)} \sum_{j \neq i} \frac{\mathbf{UCB}_{ij}(t) - p_{ij}}{2} \mathbf{1}(\mathcal{E}_{ij}(t)) \right] \\
\leq{} & \textstyle\sum_{t=t_0+1}^{T} \mathbb{E}\left[ \mathbb{E}\left[ \sum_{i \in \mathcal{I}(t)} \sum_{j \neq i} \mathbb{P}[\mathcal{E}_{ij}(t)] \frac{\overline{\mathbf{UCB}}}{2} \Big| \mathcal{I}(t) \right] \right] \\
\overset{(d)}{\leq}{} & \textstyle\sum_{t=t_0+1}^{T} \mathbb{E}\left[ \mathbb{E}\left[ \sum_{i \in \mathcal{I}(t)} \sum_{j \neq i} \frac{1}{t^{2\alpha}} \cdot \frac{\overline{\mathbf{UCB}}}{2} \Big| \mathcal{I}(t) \right] \right] \\
\leq{} & (K-1)\overline{\mathbf{UCB}} \sum_{t=t_0+1}^{\infty} t^{-2\alpha}.
\end{aligned}
\tag{39}
$$

Inequality (d) holds due to Definition 2 and Proposition 1.

Bound $\sum_{t=t_0+1}^{T} \mathbb{E}[\Phi_{1,2}(t)]$: It is straightforward that $\sum_{t=t_0+1}^{T} \mathbb{E}[\Phi_{1,2}(t)] = \mathbb{E}[\sum_{t=t_0+1}^{T} \Phi_{1,2}(t)]$. According to the definition of event $\overline{\mathcal{E}}_{ij}(t)$, we have

$$
\begin{aligned}
& \textstyle\sum_{t=t_0+1}^{T} \Phi_{1,2}(t) \\
\overset{(e)}{\leq}{} & \textstyle\sum_{t=t_0+1}^{T} \sum_{i \in \mathcal{I}(t)} \sum_{j \neq i} r_{ij}(t) \overline{\mathbf{UCB}} \\
\leq{} & \sqrt{\alpha \log(T)} \sum_{t=t_0+1}^{T} Z(t) \overline{\mathbf{UCB}}.
\end{aligned}
\tag{40}
$$

where $Z(t)$ is equal to

$$\sum_{i \in \mathcal{I}(t)} \sum_{j \neq i} \sum_{b \in \mathcal{B}_{ij}(t)} \frac{|2\eta_b^{\mathrm{A}} - 1|}{\sum_{b' \in \mathcal{B}_{ij}(t)} (2\eta_{b'}^{\mathrm{A}} - 1)^2 \sqrt{(N_{ij}^b(t) + N_{ji}^b(t))}}. \tag{41}$$

Inequality (e) holds because event $\overline{\mathcal{E}}_{ij}(t)$ implies $\text{UCB}_{ij}(t) - p_{ij} \leq 2r_{ij}(t)$. Let $\Gamma \triangleq \sum_{b' \in \mathcal{B}_{ij}(t)} (2\eta_{b'}^{\mathrm{A}} - 1)^2 / |\mathcal{B}_{ij}(t)|$. We now have $\sum_{t=t_0+1}^{T} Z(t)$:

$$\sum_{t=t_0+1}^{T} Z(t) = \sum_{t=t_0+1}^{T} \sum_{i \in \mathcal{K}} \sum_{j \neq i} \frac{1}{\Gamma |\mathcal{B}_{ij}(t)|} \sum_{b \in \mathcal{B}_{ij}(t)} \frac{\mathbf{1}(i \in \mathcal{I}(t)) |2\eta_b^A - 1|}{\sqrt{N_{ij}^b(t) + N_{ji}^b(t)}} \tag{42}$$

In Appendix F.2, we prove that the upper bound of $\sum_{t=t_0+1}^{T} \mathbb{E}[Z(t)]$ is given as follows:

$$\sum_{t=t_0+1}^{T} \mathbb{E}[Z(t)] \leq \frac{2(K-1)}{\Gamma} H + \frac{2(K-1)}{\Gamma} B^2 \log(BT^{2\alpha}) + \frac{2\sqrt{2B}}{\Gamma} \sqrt{K(K-1)T}, \tag{43}$$

where $H = \sum_{t=t_0+1}^{\infty} t^{-2\alpha}$. By substituting (43) into (40),

$$\mathbb{E}\left[ \sum_{t=t_0+1}^{T} \Phi_{1,2}(t) \right] \leq \frac{2(K-1)\overline{\text{UCB}}\sqrt{\alpha \log(T)}}{\Gamma} \left( H + B^2 \log(BT^{2\alpha}) + \sqrt{\frac{2BKT}{K-1}} \right). \tag{44}$$

To sum up, by substituting (35), (39), and (44), we can determine the bound of $\sum_{t=1}^{T} \mathbb{E}[\Phi_1(t)]$:

$$\frac{1}{K-1} \sum_{t=1}^{T} \mathbb{E}\left[\Phi_1(t)\right] \leq \overline{\text{UCB}}(t_0 + H)$$

$$+ \frac{2\overline{\text{UCB}}\sqrt{\alpha \log(T)}}{\Gamma} \left( H + B^2 \log(BT^{2\alpha}) + \sqrt{\frac{2BKT}{K-1}} \right). \tag{45}$$

**Bound** $\sum_{t=1}^{T} \mathbb{E}[\Phi_2(t)]$**:** According to the definition of $\Phi_2(t)$, we can determine the following inequality:

$$\Phi_2(t) \leq \sum_{j \neq i^*} \frac{p_{i^*j} - \text{UCB}_{i^*j}(t)}{2} \mathbf{1}(p_{i^*j} \geq \text{UCB}_{i^*j}(t)) \tag{46}$$

Hence, considering the iteration before and after $t_0$ rounds,

$$
\begin{aligned}
\sum_{t=1}^{T} \mathbb{E}[\Phi_2(t)] \\
\leq \sum_{t=1}^{t_0} \sum_{j \neq i^*} \frac{\overline{\text{UCB}}}{2} + \sum_{t=t_0+1}^{T} \sum_{j \neq i^*} \frac{\overline{\text{UCB}}}{2} \cdot \frac{1}{t^{2\alpha}} \\
\leq \frac{K-1}{2} \left[ \overline{\text{UCB}} t_0 + \overline{\text{UCB}} \sum_{t=t_0+1}^{\infty} t^{-2\alpha} \right].
\end{aligned}
\tag{47}
$$

Substituting (45) and (47) completes the proof.

## F.2  Upper Bound of $\sum_{t=t_0+1}^{T} \mathbb{E}[Z(t)]$

We now derive the bound of $\sum_{t=t_0+1}^{T} \mathbb{E}[Z(t)]$. Recall that

$$\sum_{t=t_0+1}^{T} Z(t) = \sum_{t=t_0+1}^{T} \sum_{i \in \mathcal{K}} \sum_{j \neq i} \frac{1}{\Gamma |\mathcal{B}_{ij}(t)|} \sum_{b \in \mathcal{B}_{ij}(t)} \frac{\mathbf{1}(i \in \mathcal{I}(t)) |2\eta_b^A - 1|}{\sqrt{N_{ij}^b(t) + N_{ji}^b(t)}} \tag{48}$$

The evaluation selection frequency $\mathbf{n}_{ij}(t) \triangleq (n_{ij}^b(t))_{b \in \mathcal{B}}$ satisfies the total frequency constraint $\sum_{b \in \mathcal{B}} n_{ij}^b(t) = N_{ij}(t)$. Recall that $B = |\mathcal{B}|$. Define the event

$$X_{ij}(N, t) \triangleq \left\{ \text{there exists } b \in \mathcal{B} \text{ s.t. } n_{ij}^b - \frac{N}{B} < -\sqrt{\frac{N \log(Bt^{2\alpha})}{2}} \text{ and } \sum_{b \in \mathcal{B}} n_{ij}^b = N \right\}, \tag{49}$$

and let $X_{ij}^c(N, t)$ be its complement. Since we suppose $\mathbf{n}_{ij}(t)$ follows the uniform multinominal distribution with total selection number of $N_{ij}(t)$, $n_{ij}^b(t)$ follows the binominal distribution with probability $1/B$. Applying Hoeffding Inequality, this leads to

$$\mathbf{Pr}(n_{ij}^b(t) - N_{ij}(t)/B < -\epsilon) \leq \exp\left(-\frac{2\epsilon^2}{N_{ij}(t)}\right), \tag{50}$$

for any $b \in \mathcal{B}$ and for any $\epsilon > 0$. By taking $\epsilon = \sqrt{N_{ij}(t)\log(Bt^{2\alpha})/2}$, we obtain

$$\mathbf{Pr}(X_{ij}(N_{ij}(t), t)) \leq B \cdot \exp(-\log(Bt^{2\alpha})) \leq t^{-2\alpha}. \tag{51}$$

Recall that $n_{ij}^b(t) = N_{ij}^b(t) + N_{ji}^b(t)$. Thus,

$$
\sum_{t=t_0+1}^{T} Z(t) = \sum_{t=t_0+1}^{T} \sum_{i \in \mathcal{K}} \sum_{j \neq i} \frac{\mathbf{1}(X_{ij}(N_{ij}(t), t)) + \mathbf{1}(X_{ij}^c(N_{ij}(t), t))}{\Gamma|\mathcal{B}_{ij}|} \sum_{b \in \mathcal{B}_{ij}(t)} \frac{\mathbf{1}(i \in \mathcal{I}(t))|2\eta_b^A - 1|}{\sqrt{n_{ij}^b(t)}}
$$

$$
\leq \underbrace{\left( \sum_{t=t_0+1}^{T} \sum_{i \in \mathcal{K}} \sum_{j \neq i} \frac{\mathbf{1}(i \in \mathcal{I}(t))\mathbf{1}(X_{ij}(N_{ij}(t), t))}{\Gamma} \right)}_{\sum_{t=t_0+1}^{T} Z_1(t)}
$$

$$
+ \underbrace{\left( \sum_{t=t_0+1}^{T} \sum_{i \in \mathcal{K}} \sum_{j \neq i} \frac{\mathbf{1}(i \in \mathcal{I}(t))\mathbf{1}(X_{ij}^c(N_{ij}(t), t))}{\Gamma|\mathcal{B}_{ij}|} \sum_{b \in \mathcal{B}_{ij}(t)} \frac{|2\eta_b^A - 1|}{\sqrt{n_{ij}^b(t)}} \right)}_{\sum_{t=t_0+1}^{T} Z_2(t)}.
$$

$$\tag{52}$$

For the first item on $\sum_{t=t_0+1}^{T} Z_1(t)$,

$$
\sum_{t=t_0+1}^{T} \mathbb{E}\left[Z_1(t)\right] = \frac{2}{\Gamma} \sum_{t=t_0+1}^{T} \sum_{i<j} \mathbf{1}(i \in \mathcal{I}(t))\mathbf{Pr}(X_{ij}(N_{ij}(t), t))
$$

$$
\leq \frac{2(K-1)}{\Gamma} \sum_{t=t_0+1}^{T} t^{-2\alpha}. \tag{53}
$$

For the second item on $\sum_{t=t_0+1}^{T} Z_2(t)$, we define $m(t) \triangleq B^2 \log(Bt^{2\alpha})$. For $n \geq m(t)$, we have $\frac{n}{B} - \sqrt{\frac{n\log(Bt^{2\alpha})}{2}} \geq \frac{n}{2B} \geq 1$. Then, we can conclude that

$$
\sum_{t=t_0+1}^{T} Z_2(t) \leq \frac{2}{\Gamma} \sum_{i<j} \mathbf{1}(i \in \mathcal{I}(t)) \sum_{n=1}^{T} \frac{1}{|\mathcal{B}_{ij}|} \sum_{b \in \mathcal{B}_{ij}} \frac{\mathbf{1}(X_{ij}^c(n, T))}{\sqrt{\tilde{n}_{ij}^b(n)}}
$$

$$
\leq \frac{2}{\Gamma} \sum_{i<j} \mathbf{1}(i \in \mathcal{I}(t)) \sum_{n=1}^{N_{ij}(T)} f(n, T),
$$

$$
\leq \frac{2}{\Gamma} \sum_{i<j} \mathbf{1}(i \in \mathcal{I}(t)) \left( m(T) + \sum_{n=m(t)}^{N_{ij}(T)} f(n, T) \right) \tag{54}
$$

$$
\leq \frac{2}{\Gamma} \sum_{i<j} \left( \mathbf{1}(i \in \mathcal{I}(t))B^2 \log(BT^{2\alpha}) + \sum_{n=m(T)}^{N_{ij}(T)} f(n, T) \right),
$$

where $\tilde{n}_{ij}^b(n)$ is the number of times that evaluators with bias $b$ participate among these $n$ selections. Meanwhile, $f(n, t)$ for $n \geq B$ is defined as follows:

$$
f(n, t) := \max_{\{x_i\}_{i=1}^{B}, B} \frac{1}{B} \sum_{i=1}^{B} \frac{1}{\sqrt{x_i}}
$$

$$
\text{s.t. } \sum_{i=1}^{B} x_i = n, \tag{55}
$$

$$
x_i \geq \frac{n}{B} - \sqrt{\frac{n\log(Bt^{2\alpha})}{2}}, \ i \in 1, \ldots, B,
$$

$$
B \in \{1, 2, \ldots, \min\{n, |\mathcal{B}|\}\}.
$$

We define an auxiliary function $g(n, t, B)$ as follows:

$$g(n,t,B) := \max_{\{x_i\}_{i=1}^{B}, B} \frac{1}{B} \sum_{i=1}^{B} \frac{1}{\sqrt{x_i}}$$

$$\text{s.t.} \sum_{i=1}^{B} x_i = n, \tag{56}$$

$$x_i \geq \frac{n}{B} - \sqrt{\frac{n \log(Bt^{2\alpha})}{2}}, \; i \in 1, \ldots, B.$$

It is easy to show that $f(n,t) = g(n, t, \min(n, B)) \leq g(n, t, B) \leq \sqrt{\frac{2B}{n}}$. This leads to

$$\sum_{n=m(t)}^{N_{ij}(T)} f(n,t) \leq \sqrt{2B} \sum_{n=1}^{N_{ij}(T)} \frac{1}{\sqrt{n}} \leq \sqrt{2BN_{ij}(T)}. \tag{57}$$

Hence,

$$\sum_{t=t_0+1}^{N_{ij}(T)} Z_2(t) \leq \frac{2}{\Gamma} \sum_{i<j} \left( \mathbf{1}(i \in \mathcal{I}(t)) B^2 \log(BT^{2\alpha}) + \sqrt{2BN_{ij}(T)} \right)$$

$$\leq \frac{2(K-1)}{\Gamma} B^2 \log(BT^{2\alpha}) + \frac{2\sqrt{2B}}{\Gamma} \sqrt{K(K-1) \sum_{i<j} N_{ij}(T)} \tag{58}$$

$$= \frac{2(K-1)}{\Gamma} B^2 \log(BT^{2\alpha}) + \frac{2\sqrt{2B}}{\Gamma} \sqrt{K(K-1)T}.$$

Combining $Z_1(t)$ and $Z_2(t)$, we get

$$\sum_{t=t_0+1}^{T} \mathbb{E}[Z(t)] \leq \frac{2(K-1)}{\Gamma} H + \frac{2(K-1)}{\Gamma} B^2 \log(BT^{2\alpha}) + \frac{2\sqrt{2B}}{\Gamma} \sqrt{K(K-1)T}, \tag{59}$$

where $H = \sum_{t=t_0+1}^{\infty} t^{-2\alpha}$.

## Appendix G    Extended Bias-Sensitive UCB Algorithm

Algorithm 2 presents the extended bias-sensitive UCB algorithm for unknown bias case.

---
**Algorithm 2** Extended Bias-Sensitive UCB Algorithm

---
1:  **for** each time slot $t = 1$ to $T$ **do**
2:      Set $\hat{p}_{ij}(t-1) = \bar{p}_{ij}^m(t-1)$, compute $\hat{\eta}_m(t-1)$ using (18);
3:      **Repeat** twice
4:          Update $\hat{p}_{ij}(t-1)$ using (16), update $\hat{\eta}_m(t-1)$ using (18);
5:      Compute $\hat{r}_{ij}(t-1)$ using (19);
6:      Substitute $\hat{p}_{ij}(t-1)$ and $\hat{r}_{ij}(t-1)$ into (12) and (13) to compute $\text{UCB}_i(t), i \in \mathcal{K}$;
7:      Select the arms $x_1(t)$ and $x_2(t)$ that optimize problem (14);
8: **end for**

---

As we mentioned in the main context, line 2 and the first update of $\hat{p}_{ij}(t-1)$ and $\hat{\eta}_m(t-1)$ in line 4 can be skipped after a certain number of time slots once the estimation tends to be relatively accurate to accelerate the convergence. We empirically find that this time slot threshold can be set in the form of $cK \log K$. We set $c = 50$ in experiments. The explanation of the twice update in lines 3–4 can be found in Appendix K.7.

## Appendix H  Proof for Lemma 3

The proof path is similar as that for Proposition 1. To alleviate the coupling of the randomness among requests, we introduce a $M \times t$ table. Cell $(m \in \mathcal{M}, s \in \mathcal{T})$ corresponds to the $s$-th times that $\{i, j\}$ is selected for evaluator $m$. Let $N_{ij}^{m,s}$ denote the number of time slots $t'$ (i) until (i.e., on and before) the $s$-th times that $\{i, j\}$ is selected for evaluator $m$ and (ii) evaluator $m$ sends a feedback with $o_i \succ_m o_j$. Let $\bar{q}_{i,j}^{m,s}$ denote the fraction of feedback sent by evaluator $m$ claiming $o_i \succ_m o_j$ among the first $s$ times that $\{i, j\}$ is selected for that evaluator. That is,

$$\bar{q}_{ij}^{m,s} = \frac{N_{ij}^{m,s}}{s}. \tag{60}$$

Then, according to Hoeffding Inequality, for any $t > 0$,

$$\Pr\left( |\bar{q}_{ij}^{m,s} - \mathbb{E}[\bar{q}_{ij}^{m,s}]| \le \sqrt{\frac{\alpha \log(t)}{s}} \right) \ge 1 - \frac{2}{t^{2\alpha}}, \tag{61}$$

where $\mathbb{E}[\bar{q}_{ij}^{m,s}]$ denotes the expected value of $\bar{q}_{ij}^{m,s}$ considering the randomness of feedback due to arm winning probability and evaluator bias. According to (4), $\mathbb{E}[\bar{q}_{ij}^{m,s}]$ can be determined by

$$\mathbb{E}[\bar{q}_{ij}^{m,s}] = \eta_m p_{ij} + (1 - \eta_m)(1 - p_{ij}). \tag{62}$$

Substituting (62) into the inequality on the left-hand side of (61), we have

$$|\bar{q}_{ij}^{m,s} - (\eta_m p_{ij} + (1 - \eta_m)(1 - p_{ij}))| \le \sqrt{\frac{\alpha \log(t)}{s}}. \tag{63}$$

Then, multiplying both sides by $|2\hat{\eta}_m(t) - 1|$, the following inequality holds for all $m \in \mathcal{M}$ and $s \in \mathcal{T}$:

$$| \left( N_{ij}^{m,s}/s - (1 - \eta_m) \right) (2\hat{\eta}_m(t) - 1) - (2\eta_m - 1)(2\hat{\eta}_m(t) - 1)p_{ij} | \le |2\hat{\eta}_m(t) - 1| \sqrt{\frac{\alpha \log(t)}{s}}. \tag{64}$$

Let $s = N_{ij}^m(t) + N_{ji}^m(t)$ and hence $N_{ij}^{m,s} = N_{ij}^m(t)$. Then, we reorganize the inequality expression inside function $\Pr(\cdot)$. By extracting $\epsilon_m^\eta(t) \triangleq (2\eta_m - 1)/(2\hat{\eta}_m(t) - 1)$ at the left-hand side of the inequality and substituting $\phi_{ij}^m(t) \triangleq (N_{ij}^m(t)/(N_{ij}^m(t) + N_{ji}^m(t)) - (1 - \eta_m))(2\eta_m - 1)$ and $\hat{\phi}_{ij}^m(t) \triangleq (N_{ij}^m(t)/(N_{ij}^m(t) + N_{ji}^m(t)) - (1 - \hat{\eta}_m(t)))(2\hat{\eta}_m(t) - 1)$, we have the following:

$$|\epsilon_m^\eta(t)| \, |\hat{\phi}_{ij}^m(t) - p_{ij}(2\hat{\eta}_m(t) - 1)^2 + (\phi_{ij}^m(t)/(\epsilon_m^\eta(t))^2 - \hat{\phi}_{ij}^m(t))| \le |2\hat{\eta}_m(t) - 1| \sqrt{\frac{\alpha \log(t)}{s}}. \tag{65}$$

By triangle inequality and rearranging the inequality expression, we can obtain the following inequality:

$$\left| \hat{\phi}_{ij}^m(t) - p_{ij}(2\hat{\eta}_m(t) - 1)^2 \right| \le \left| \phi_{ij}^m(t)/(\epsilon_m^\eta(t))^2 - \hat{\phi}_{ij}^m(t) \right| + \frac{|2\hat{\eta}_m(t) - 1|}{|\epsilon_m^\eta(t)|} \sqrt{\frac{\alpha \log(t)}{N_{ij}^m(t) + N_{ji}^m(t)}}. \tag{66}$$

Based on triangle inequality, considering (66) for all possible $m \in \mathcal{M}_{ij}(t)$, the following inequality holds:

$$\Pr\left( \left| \sum_{m \in \mathcal{M}_{ij}(t)} \left( \frac{N_{ij}^m(t)}{N_{ij}^m(t) + N_{ji}^m(t)} - (1 - \hat{\eta}_m(t)) \right) (2\hat{\eta}_m(t) - 1) - \sum_{m \in \mathcal{M}_{ij}(t)} p_{ij}(2\hat{\eta}_m(t) - 1)^2 \right| \right.$$
$$\left. \le \sum_{m \in \mathcal{M}_{ij}(t)} \left| \phi_{ij}^m(t)/(\epsilon_m^\eta(t))^2 - \hat{\phi}_{ij}^m(t) \right| + \sum_{m \in \mathcal{M}_{ij}(t)} \frac{|2\hat{\eta}_m(t) - 1|}{|\epsilon_m^\eta(t)|} \sqrt{\frac{\alpha \log(t)}{N_{ij}^m(t) + N_{ji}^m(t)}} \right) \ge 1 - \frac{2}{t^{2\alpha}}. \tag{67}$$

Dividing both sides of the inequality in $\Pr(\cdot)$ by factor $\sum_{m \in \mathcal{M}_{ij}(t)} (2\hat{\eta}_m(t) - 1)^2$, we complete the proof.

## Appendix I   Proof for Theorem 2

The proof path is exactly same as that in Appendix F except that the event probability $\mathbb{P}[\mathcal{E}_{ij}(t)]$ in (39) and $\mathbb{P}[p_{i^*j} \geq \text{UCB}_{i^*j}(t)]$ in (47) are changed. Specifically, as in Appendix F, for any request $t = t_0 + 1, ..., T$, there are two possible events:

- $\mathcal{E}_{ij}(t)$: $\hat{p}_{ij}(t) - p_{ij} > \hat{r}_{ij}(t)$;
- $\overline{\mathcal{E}}_{ij}(t)$: complement of $\mathcal{E}_{ij}(t)$.

According to Definition 3,

$$\mathbb{P}[\mathcal{E}_{ij}(t)] = \frac{1 - \mathbb{P}[|\hat{p}_{ij}(t) - p_{ij}| \leq \hat{r}_{ij}(t)]}{2} \leq \frac{1 - \mathbb{P}[|\hat{p}_{ij}(t) - p_{ij}| \leq r_{ij}^{\circ}(t)]}{2} + \xi(t). \quad (68)$$

By Lemma 3, $(1 - \mathbb{P}[|\hat{p}_{ij}(t) - p_{ij}| \leq r_{ij}^{\circ}(t)])/2 + \xi(t) \leq 1/t^{2\alpha} + \xi(t)$ and hence $\mathbb{P}[\mathcal{E}_{ij}(t)] \leq 1/t^{2\alpha} + \xi(t)$. Due to the same reason, $\mathbb{P}[p_{i^*j} \geq \text{UCB}_{i^*j}(t)] \leq 1/t^{2\alpha} + \xi(t)$. Substituting $\mathbb{P}[\mathcal{E}_{ij}(t)]$ in (39) and $\mathbb{P}[p_{i^*j} \geq \text{UCB}_{i^*j}(t)]$ in (47), we complete the proof.

## Appendix J   Discussion on $\xi(t)$

Let $A(T) = \frac{1}{T} \sum_{t=t_0+1}^{T} \xi(t)$. Proving $\lim_{T \to \infty} A(T) \to 0$ can be challenging due to the non-convex joint bias and winning probability estimation problem in (17). Instead, we provide the following evidences to show that $A(T)$ can be small and may approach zero.

(i) As $t \to \infty$, we show that $\hat{\eta}_m(t) \to \eta_m + \epsilon$. Showing this result analytically is challenging. This is because due to the non-convexity, proving such a performance guarantee under BCD remains an open problem. We empirically show the convergence of the bias estimation error $|\eta_m(t) - \eta_m|$ in Appendix K.3.

(ii) As $t \to \infty$, if $\hat{\eta}_m(t) \to \eta_m + \epsilon$, then we show that $\xi(t)$ is non-increasing and converge to value $\epsilon_\xi$, so $\lim_{T \to \infty} A(T) \to \epsilon_\xi$. Here, $\epsilon_\xi$ decreases as $|\epsilon|$ decreases, and it approaches zero if $\epsilon \to 0$. Specifically, recall that parameter $\xi(t)$ is the minimum non-negative value such that $\xi(t) \geq \frac{1}{2} \left( \text{HF} \left( \mathbb{P}\left( |\hat{p}_{ij}(t) - p_{ij}| \leq r_{ij}^{\circ}(t) \right) \right) - \mathbb{P}\left( |\hat{p}_{ij}(t) - p_{ij}| \leq \hat{r}_{ij}(t) \right) \right)$. There are two possible cases: (a) $r_{ij}^{\circ}(t) \leq \hat{r}_{ij}(t)$, under which $\xi(t) = 0$ always holds; (b) $r_{ij}^{\circ}(t) > \hat{r}_{ij}(t)$, under which $\xi(t) > 0$. We can show that if $\hat{\eta}_m(t) \to \eta_m + \epsilon$, then based on the definition of confidence radius, the gap between $r_{ij}^{\circ}(t)$ and $\hat{r}_{ij}(t)$ changes continuously, so the analysis can focus on case (b). Based on Hoeffding inequality and the definition of $\xi(t)$,

$$\mathbb{P}\left( |\hat{p}_{ij}(t) - p_{ij}| \leq r_{ij}^{\circ}(t) \right) \geq 1 - 2/t^{2\alpha}, \quad (69)$$

$$\mathbb{P}\left( |\hat{p}_{ij}(t) - p_{ij}| \leq \hat{r}_{ij}(t) \right) \geq 1 - 2/t^{2\alpha} + \xi(t). \quad (70)$$

Based on the definition of confidence radius, as $\hat{\eta}_m(t) \to \eta_m + \epsilon$, there exists a sequence $\hat{\alpha}(t)$ such that $\hat{r}_{ij}(t; \hat{\alpha}(t)) = r_{ij}^{o}(t)$ with $\hat{\alpha}(t)$ is non-decreasing and approaches $\alpha$. Here, $\hat{r}_{ij}(t; \hat{\alpha}(t))$ is the formulation of $\hat{r}_{ij}(t)$ with $\alpha = \hat{\alpha}(t)$ substituted. Consequently, as $\hat{\alpha}(t) \to \alpha$, $\xi(t)$ is non-increasing and approaches to $\epsilon_\xi$ as $t \to \infty$. On the other hand, we would like to clarify that it is almost impossible to empirically show the convergence of $\xi(t)$, because $\xi(t)$ is defined based on the probability that the estimation falls within a certain range (rather than a deterministic value) and hence difficult to compute in practice.

Summing up evidences (i) and (ii), then as $t \to \infty$, we have $\xi(t) \to \epsilon_\xi$ and $\lim_{T \to \infty} A(T) \to \epsilon_\xi$, i.e., $A(T)$ can be small and possibly as small as zero when $T$ is sufficiently large.

## Appendix K   Additional Experiments

### K.1   Incorporating Our Estimators into Baselines

In this section, we describe how to incorporate our estimators into RC, RUCB, and DT and obtain their bias-sensitive versions RC-B, RUCB-B and DT-B. The modification ideas are similar: (i) introduce a 3-dimensional (3D) array $\mathbf{V}$ to record the choices of multiple evaluators; (ii) replace the

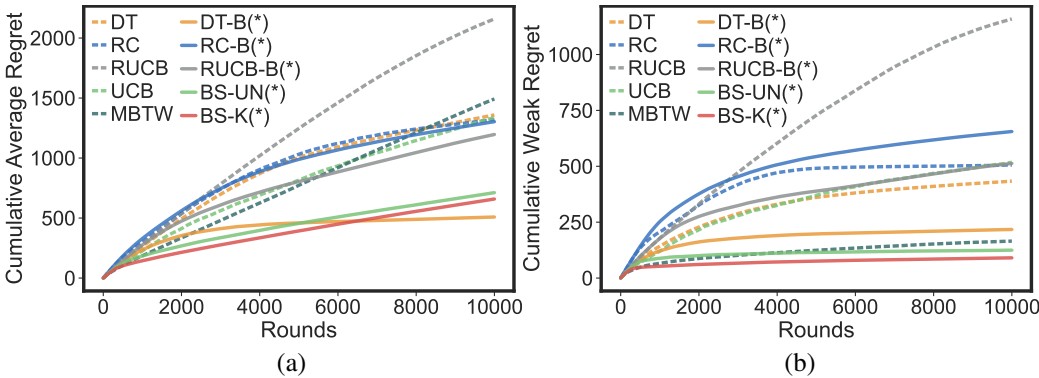

Figure 1: Algorithm convergence: (a) cumulative average regret, and (b) cumulative weak regret.

arm performance estimator in baselines with our estimators presented in 2–5 of our Algorithm 2; (iii) map the result back to the case without the definition of $\mathbf{V}$ such that the other steps in the original baselines remain applicable. The details of the bias-sensitive versions RC-B, RUCB-B, and DT-B are given as follows, where their main differences rely on which line to modify and what notations to use.

**RC-B:** Modify Algorithm 1 in [10]:

- Line 2: Initialize a 3D array $\mathbf{V} \leftarrow \mathbf{0}_{K \times K \times M}$, where $M$ is the number of evaluators;

- Line 12: Replace it with lines 2–5 in our Algorithm 2, and set $U_{ij}(t) = p_{ij} + r_{ij}$;

- Line 16: Replace it with $V_{cd}^k \leftarrow V_{cd}^k + 1$ if $c \succ_k d$, and let $W_{ij} = \sum_{k \in \mathcal{K}} V_{ij}^k$.

**RUCB-B:** Modify Algorithm 1 in [11]:

- Line 1: Initialize a 3D array $\mathbf{V} \leftarrow \mathbf{0}_{K \times K \times M}$, where $M$ is the number of evaluators;

- Line 4: Replace it with lines 2–5 in our Algorithm 2, and set $u_{ij} = p_{ij} + r_{ij}$;

- Line 14: Replace it with $V_{cd}^k \leftarrow V_{cd}^k + 1$ if $c \succ_k d$, and let $W_{ij} = \sum_{k \in \mathcal{K}} V_{ij}^k$.

**DT-B:** Modify Algorithm 1 in [12]:

- Line 1: Initialize a 3D array $\mathbf{V} \leftarrow \mathbf{0}_{K \times K \times M}$, where $M$ is the number of evaluators;

- Line 4: Replace it with lines 2–5 in our Algorithm 2, and set $u_{ij} = p_{ij} + r_{ij}$ and $l_{ij} = p_{ij} - r_{ij}$;

- Line 17: Replace it with $V_{ij}^k \leftarrow V_{ij}^k + 1$ if $i \succ_k j$, and let $B_{ij} = \sum_{k \in \mathcal{K}} V_{ij}^k$.

### K.2 Algorithm Convergence and Standard Error

Figure 1 shows the convergence of average and weak regrets of our methods and other baselines. We have the following observations. First, our proposed BS-UN and BS-K methods have much lower average and weak regrets than baselines (marked with dashed line). When $t = 10000$, the reduction of average regret is more than 40%. Second, when compared with the baselines (marked with dashed lines), their bias-sensitive versions (marked with solid lines) achieve slower regret increase, verifying the capability of the bias estimation technique of our methods. This reduction is especially significant for DT and RUCB. Third, the performance of BS-UN and BS-K are similar, indicating the effectiveness of the bias estimation proposed in our methods.

Tables 6 and 7 show the results of Table 1 with standard error. Tables 8 and 9 show those of Table 2. Here, the standard error is defined as the standard deviation divided by the number of experiments (i.e., 100 in this work).

| | Arm Heter. $\sigma^2$ | | | Bias Concentr. $\alpha_{\text{B}}$ | | |
|---|---|---|---|---|---|---|
| | 1.0 | 2.0 | 4.0 | 1.0 | 2.0 | 3.0 |
| RC | $1374_{\pm1.69\%}$ | $1338_{\pm1.39\%}$ | $967_{\pm0.91\%}$ | $2845_{\pm0.16\%}$ | $1338_{\pm1.39\%}$ | $687_{\pm1.38\%}$ |
| RUCB | $1906_{\pm0.43\%}$ | $2134_{\pm0.70\%}$ | $1154_{\pm0.27\%}$ | $2832_{\pm0.17\%}$ | $2134_{\pm0.70\%}$ | $1185_{\pm0.93\%}$ |
| DT | $1396_{\pm2.21\%}$ | $1425_{\pm2.59\%}$ | $942_{\pm1.50\%}$ | $2621_{\pm0.68\%}$ | $1425_{\pm2.59\%}$ | $640_{\pm3.14\%}$ |
| MBTW | $1220_{\pm1.92\%}$ | $1509_{\pm1.59\%}$ | $726_{\pm1.91\%}$ | $1769_{\pm2.10\%}$ | $1509_{\pm1.59\%}$ | $1448_{\pm1.72\%}$ |
| UCB | $1283_{\pm2.09\%}$ | $1426_{\pm2.46\%}$ | $732_{\pm2.46\%}$ | $2581_{\pm0.99\%}$ | $1426_{\pm2.46\%}$ | $706_{\pm3.20\%}$ |
| **RC-B** (*) | $1378_{\pm1.54\%}$ | $1611_{\pm1.63\%}$ | $1050_{\pm0.88\%}$ | $2207_{\pm1.48\%}$ | $1611_{\pm1.63\%}$ | $803_{\pm2.00\%}$ |
| **RUCB-B** (*) | $993_{\pm0.58\%}$ | $1120_{\pm0.47\%}$ | $709_{\pm0.49\%}$ | $1191_{\pm5.91\%}$ | $1120_{\pm0.47\%}$ | $1055_{\pm0.38\%}$ |
| **DT-B** (*) | $\mathbf{430}_{\pm3.99\%}$ | $\mathbf{411}_{\pm4.30\%}$ | $\mathbf{344}_{\pm3.22\%}$ | $631_{\pm14.53\%}$ | $\mathbf{411}_{\pm4.30\%}$ | $\mathbf{280}_{\pm4.13\%}$ |
| **BS-UN** (*) | $690_{\pm3.63\%}$ | $689_{\pm5.98\%}$ | $387_{\pm2.15\%}$ | $825_{\pm11.41\%}$ | $689_{\pm5.98\%}$ | $637_{\pm3.98\%}$ |
| **BS-K** (*) | $654_{\pm4.29\%}$ | $713_{\pm4.66\%}$ | $407_{\pm3.82\%}$ | $\mathbf{554}_{\pm3.21\%}$ | $713_{\pm4.66\%}$ | $624_{\pm4.25\%}$ |

Table 6: Cumulative average regret ($\downarrow$) in Table 1 with standard error.

| | Arm Heter. $\sigma^2$ | | | Bias Concentr. $\alpha_{\text{B}}$ | | |
|---|---|---|---|---|---|---|
| | 1.0 | 2.0 | 4.0 | 1.0 | 2.0 | 3.0 |
| RC | $596_{\pm2.94\%}$ | $525_{\pm2.46\%}$ | $502_{\pm1.85\%}$ | $1847_{\pm0.32\%}$ | $525_{\pm2.46\%}$ | $278_{\pm1.75\%}$ |
| RUCB | $1018_{\pm0.70\%}$ | $1144_{\pm1.10\%}$ | $719_{\pm0.53\%}$ | $1829_{\pm0.35\%}$ | $1144_{\pm1.10\%}$ | $506_{\pm1.42\%}$ |
| DT | $445_{\pm5.12\%}$ | $492_{\pm5.45\%}$ | $375_{\pm4.02\%}$ | $1428_{\pm2.09\%}$ | $492_{\pm5.45\%}$ | $191_{\pm6.52\%}$ |
| MBTW | $175_{\pm11.96\%}$ | $162_{\pm12.82\%}$ | $140_{\pm12.88\%}$ | $569_{\pm8.53\%}$ | $162_{\pm12.82\%}$ | $92_{\pm16.14\%}$ |
| UCB | $553_{\pm3.19\%}$ | $548_{\pm3.35\%}$ | $336_{\pm3.39\%}$ | $1583_{\pm1.69\%}$ | $548_{\pm3.35\%}$ | $153_{\pm4.76\%}$ |
| **RC-B** (*) | $649_{\pm1.48\%}$ | $869_{\pm1.45\%}$ | $727_{\pm0.97\%}$ | $1119_{\pm3.35\%}$ | $869_{\pm1.45\%}$ | $502_{\pm1.83\%}$ |
| **RUCB-B** (*) | $422_{\pm0.87\%}$ | $480_{\pm0.65\%}$ | $370_{\pm0.56\%}$ | $604_{\pm12.35\%}$ | $480_{\pm0.65\%}$ | $446_{\pm0.54\%}$ |
| **DT-B** (*) | $198_{\pm6.93\%}$ | $210_{\pm7.55\%}$ | $168_{\pm4.77\%}$ | $436_{\pm21.20\%}$ | $210_{\pm7.55\%}$ | $110_{\pm7.59\%}$ |
| **BS-UN** (*) | $194_{\pm8.72\%}$ | $161_{\pm24.46\%}$ | $94_{\pm4.01\%}$ | $340_{\pm29.14\%}$ | $161_{\pm24.46\%}$ | $92_{\pm10.05\%}$ |
| **BS-K** (*) | $\mathbf{116}_{\pm9.26\%}$ | $\mathbf{90}_{\pm14.02\%}$ | $\mathbf{79}_{\pm8.36\%}$ | $\mathbf{60}_{\pm6.39\%}$ | $\mathbf{90}_{\pm14.02\%}$ | $\mathbf{82}_{\pm11.70\%}$ |

Table 7: Cumulative weak regret ($\downarrow$) in Table 1 with standard error.

## K.3  Bias Estimation Error

Figure 2 shows the convergence of the bias estimation error $|\hat{\eta}_m(t) - \eta_m|$ within 40000 rounds for our methods, including our proposed BS-UN and the bias-sensitive versions of baselines RC-B, RUCB-B, and DT-B. The following table shows the bias estimation error after 40000 rounds:

| (Num. of Arms, Num. of Evaluators) | (10,10) | (20,20) | (50,50) |
|---|---|---|---|
| RC-B (*) | 0.03 | 0.02 | 0.06 |
| RUCB-B (*) | 0.03 | 0.04 | 0.04 |
| DT-B (*) | 0.10 | 0.08 | 0.16 |
| BS-UN (*) | 0.04 | 0.04 | 0.03 |

Both the table and figure verify the convergence of the estimation error to small values for all our methods under diverse network scale.

## K.4  Large-Scale Settings

To enable a more stable performance under large-scale settings, we make a minor modification on Algorithm 2 such that step 2 is skipped after $\text{THR}_s = cn \log n$ iterations, where we set $c = 50$ due to its empirical performance. This is reasonable because step 2 is used for determining a suitable initialization of $\eta_m$'s. Once the $\eta_m$'s have been approximately converged, no further initialization is needed. The result is presented in Table 10. First, our proposed BS-UN achieves lower average and weak regrets than baselines under large-scale settings. Second, DT-B also has lower regrets than the baselines. This indicates that by incorporating our bias estimation technique, DT-B is able to address biased evaluator. Finally, perhaps counter-intuitive, under the large-scale settings, BS-UN outperforms BS-K. This is because the winning probability estimation at earlier iterations is relatively inaccurate, and BS-UN tends to explore more than BS-K and hence achieves better performance when the number of arms and evaluators are large.

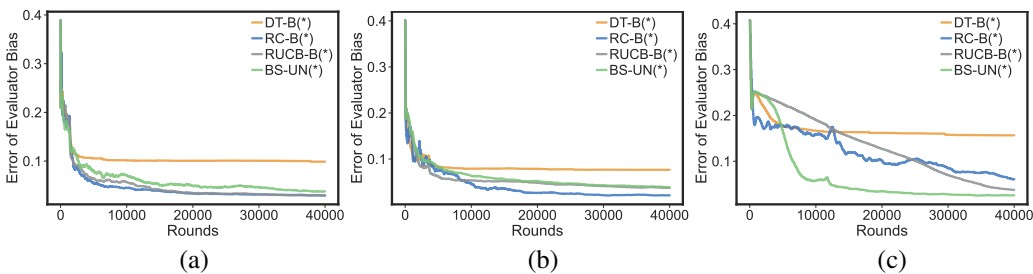

Figure 2: Bias estimation error: (a) $(10, 10)$, (b) $(20, 20)$, and (c) $(50, 50)$.

| | Number of Evaluators | | | Number of Arms | | |
|---|---|---|---|---|---|---|
| | 5 | 15 | 20 | 5 | 15 | 20 |
| RC | $1590_{\pm 1.35\%}$ | $1142_{\pm 1.50\%}$ | $1301_{\pm 1.66\%}$ | $561_{\pm 2.03\%}$ | $2250_{\pm 0.64\%}$ | $2568_{\pm 0.06\%}$ |
| RUCB | $2293_{\pm 0.49\%}$ | $1924_{\pm 0.72\%}$ | $2042_{\pm 0.75\%}$ | $813_{\pm 1.16\%}$ | $2277_{\pm 0.21\%}$ | $2499_{\pm 0.10\%}$ |
| DT | $1548_{\pm 2.29\%}$ | $1073_{\pm 3.04\%}$ | $1221_{\pm 2.42\%}$ | $695_{\pm 1.97\%}$ | $1854_{\pm 1.48\%}$ | $2006_{\pm 1.43\%}$ |
| MBTW | $1444_{\pm 1.97\%}$ | $1452_{\pm 1.87\%}$ | $1509_{\pm 1.95\%}$ | $1218_{\pm 1.95\%}$ | $1260_{\pm 1.69\%}$ | $1330_{\pm 1.77\%}$ |
| UCB | $1604_{\pm 2.18\%}$ | $1099_{\pm 3.74\%}$ | $1252_{\pm 3.30\%}$ | $631_{\pm 4.98\%}$ | $1752_{\pm 0.91\%}$ | $2115_{\pm 0.36\%}$ |
| **RC-B** (*) | $1801_{\pm 1.21\%}$ | $1370_{\pm 1.82\%}$ | $1694_{\pm 1.75\%}$ | $575_{\pm 3.38\%}$ | $2301_{\pm 0.32\%}$ | $2707_{\pm 0.06\%}$ |
| **RUCB-B** (*) | $964_{\pm 0.46\%}$ | $1102_{\pm 0.48\%}$ | $1115_{\pm 0.37\%}$ | $820_{\pm 6.95\%}$ | $1360_{\pm 0.46\%}$ | $1868_{\pm 0.37\%}$ |
| **DT-B** (*) | $\mathbf{360}_{\pm 5.22\%}$ | $\mathbf{375}_{\pm 5.51\%}$ | $\mathbf{344}_{\pm 5.17\%}$ | $\mathbf{169}_{\pm 8.38\%}$ | $\mathbf{736}_{\pm 3.09\%}$ | $\underline{1156}_{\pm 2.79\%}$ |
| **BS-UN** (*) | $688_{\pm 6.84\%}$ | $722_{\pm 4.06\%}$ | $\underline{621}_{\pm 3.24\%}$ | $626_{\pm 9.08\%}$ | $929_{\pm 1.76\%}$ | $\underline{1157}_{\pm 1.16\%}$ |
| **BS-K** (*) | $588_{\pm 4.29\%}$ | $600_{\pm 3.49\%}$ | $638_{\pm 4.55\%}$ | $469_{\pm 3.56\%}$ | $\underline{881}_{\pm 2.79\%}$ | $\mathbf{1021}_{\pm 2.28\%}$ |

Table 8: Cumulative average regret ($\downarrow$) in Table 2 with standard error.

## K.5 Additional Baselines

We have conducted experiments on Doubler [29], MultiSBM [29], MaxInP [30], and MaxMinLCB [31]. Table 11 provides the experimental results. As can be observed, our proposed BS-K and BS-UN approaches always outperform baselines in terms of both weak and average regrets.

## K.6 Ablation Study: Description and Weak Regret

We first describe the estimators we considered in ablation study. For the arm performance estimator in Figure 3 (a) and (c), we compare the following:

- Weighted-Voting-UN: estimator in (22) while replacing the actual evaluator bias with the estimated bias in (18);

- Weighted-Voting-K: estimator in (22);

- Minimum-Deviation-UN: estimator in (16)

- Minimum-Deviation-K: estimator in (10).

For the evaluator bias estimation in Figure 3 (b) and (d), we compare two groups of estimators: conditional probability expression-based estimator (denoted by "COND"); optimization-based estimator (denoted by "OPT"), which is the bias estimator in (18). For the group of COND estimators, the bias is estimated by

$$\hat{\eta}_m = \frac{\sum_{i,j}(2\hat{p}_{ij} - 1)\mathbf{1}(\hat{p}_{ij} > 0.5)\mathbf{1}(\bar{p}_{ij}^m > 0.5)}{\sum_{i,j}(2\hat{p}_{ij} - 1)\mathbf{1}(\hat{p}_{ij} > 0.5)}, \tag{71}$$

which is derived based on the definition of $\eta_m$ in (3). For either of COND and OPT estimators, calculating the estimated $\eta_m$ and $p_{ij}$ are challenging, as the calculation corresponds to solving complex equation system (e.g., the equation system comprising (16) and (18) or comprising (16) and (71)). Thus, we compare three approaches to calculate their estimated values based on our designed estimators:

| | Number of Evaluators | | | Number of Arms | | |
|---|---|---|---|---|---|---|
| | 5 | 15 | 20 | 5 | 15 | 20 |
| RC | $689_{\pm2.45\%}$ | $409_{\pm2.70\%}$ | $497_{\pm2.96\%}$ | $85_{\pm4.46\%}$ | $1303_{\pm1.07\%}$ | $1626_{\pm0.11\%}$ |
| RUCB | $1272_{\pm0.87\%}$ | $979_{\pm1.10\%}$ | $1067_{\pm1.20\%}$ | $181_{\pm2.37\%}$ | $1356_{\pm0.33\%}$ | $1593_{\pm0.17\%}$ |
| DT | $491_{\pm5.37\%}$ | $295_{\pm6.58\%}$ | $349_{\pm5.31\%}$ | $142_{\pm4.26\%}$ | $702_{\pm3.39\%}$ | $802_{\pm3.82\%}$ |
| MBTW | $177_{\pm11.53\%}$ | $124_{\pm13.78\%}$ | $182_{\pm12.35\%}$ | $48_{\pm16.45\%}$ | $307_{\pm9.63\%}$ | $438_{\pm7.83\%}$ |
| UCB | $709_{\pm3.51\%}$ | $387_{\pm5.07\%}$ | $491_{\pm4.44\%}$ | $77_{\pm15.07\%}$ | $971_{\pm1.27\%}$ | $1264_{\pm0.50\%}$ |
| **RC-B** (*) | $934_{\pm1.13\%}$ | $785_{\pm1.50\%}$ | $1005_{\pm1.70\%}$ | $198_{\pm3.77\%}$ | $1492_{\pm0.48\%}$ | $1993_{\pm0.12\%}$ |
| **RUCB-B** (*) | $392_{\pm0.70\%}$ | $468_{\pm0.60\%}$ | $483_{\pm0.61\%}$ | $243_{\pm23.70\%}$ | $707_{\pm0.77\%}$ | $1146_{\pm0.58\%}$ |
| **DT-B** (*) | $177_{\pm8.83\%}$ | $165_{\pm10.77\%}$ | $169_{\pm9.50\%}$ | $77_{\pm17.31\%}$ | $308_{\pm4.88\%}$ | $578_{\pm4.14\%}$ |
| **BS-UN** (*) | $121_{\pm35.16\%}$ | $97_{\pm6.20\%}$ | $79_{\pm3.80\%}$ | $108_{\pm52.06\%}$ | $364_{\pm4.98\%}$ | $543_{\pm2.94\%}$ |
| **BS-K** (*) | $57_{\pm9.07\%}$ | $73_{\pm9.45\%}$ | $75_{\pm11.06\%}$ | $21_{\pm19.88\%}$ | $306_{\pm7.60\%}$ | $420_{\pm5.40\%}$ |

Table 9: Cumulative weak regret ($\downarrow$) in Table 2 with standard error.

| | Cum. Weak Regret ($\downarrow$) | | Cum. Average Regret ($\downarrow$) | |
|---|---|---|---|---|
| (Num. of Arms, Num. of Eval.) | (50,50) | (100,100) | (50,50) | (100,100) |
| RC | $17494_{\pm0.01\%}$ | $18711_{\pm0.00\%}$ | $18441_{\pm0.01\%}$ | $19298_{\pm0.00\%}$ |
| RUCB | $16625_{\pm0.06\%}$ | $17887_{\pm0.02\%}$ | $17931_{\pm0.03\%}$ | $18705_{\pm0.01\%}$ |
| DT | $3634_{\pm30.11\%}$ | $7656_{\pm18.17\%}$ | $6288_{\pm15.62\%}$ | $10993_{\pm9.96\%}$ |
| MBTW | $2170_{\pm38.93\%}$ | $5725_{\pm23.24\%}$ | $10293_{\pm3.69\%}$ | $12013_{\pm5.48\%}$ |
| UCB | $15398_{\pm0.26\%}$ | $16410_{\pm0.25\%}$ | $17196_{\pm0.13\%}$ | $17853_{\pm0.12\%}$ |
| **RC-B**(*) | $18074_{\pm0.00\%}$ | $18716_{\pm0.00\%}$ | $18754_{\pm0.00\%}$ | $19313_{\pm0.00\%}$ |
| **RUCB-B**(*) | $15609_{\pm1.29\%}$ | $17871_{\pm0.06\%}$ | $17424_{\pm0.62\%}$ | $18694_{\pm0.03\%}$ |
| **DT-B**(*) | $\mathbf{1481}_{\pm11.92\%}$ | $\mathbf{2850}_{\pm9.70\%}$ | $\mathbf{1907}_{\pm10.57\%}$ | $\mathbf{3703}_{\pm8.56\%}$ |
| **BS-UN**(*) | $1628_{\pm4.94\%}$ | $5280_{\pm2.57\%}$ | $9307_{\pm0.85\%}$ | $11533_{\pm0.92\%}$ |
| **BS-K**(*) | $3825_{\pm1.93\%}$ | $17871_{\pm0.02\%}$ | $10648_{\pm0.58\%}$ | $18696_{\pm0.01\%}$ |

Table 10: Algorithm performance under large-scale settings.

- Last Round Preference (denoted by "P"): Estimate evaluators' bias based on the estimated arm performance in the previous time slot, and then update the arm performance estimation using (16);

- Mean Preference (denoted by "MP"): Estimate evaluators' bias based on $\sum_{m\in\mathcal{M}} \bar{p}_{ij}^m(t-1)/M$, and then update the arm performance estimation using (16);

- Mean Preference with User Bias (denoted by "BMP"): The approach presented in Algorithm 2.

To sum up, we compare seven approaches with different bias estimators and calculation methods: the case where the ground-truth bias is known (denoted by "Known"), which serves as the benchmark, COND-P, COND-MP, COND-BMP, OPT-P, OPT-MP, and OPT-BMP.

Now, we show the ablation study on average and weak regrets in Figure 3. Our proposed arm performance estimator leads to a much lower average and weak regrets than that in Appendix A. In addition, our proposed bias estimation approach achieves an average and weak regret that is closer to the case where the ground-truth bias is known, when compared with the other approaches.

### K.7 Ablation Study: Times in Estimation Updates

Table 12 shows the evaluation of different times of the estimation updates in lines 3–4 of Algorithm 2. Note that to show a relatively obvious performance gap, we conducted experiments under a relatively large scale setting with 50 arms and 10 evaluators.

It can be observed that "updating twice" leads to the best performance. Further increasing the times of updates does not improve the performance. Intuitively, the main idea is to find a better initial point for a more accurate estimation in each time slot. According to Algorithm 2, in line 2 of each time slot, the arm performance estimation $\hat{p}_{ij}(t-1)$ is initialized as the estimation ignoring the bias, and then the bias estimation $\hat{\eta}_m(t-1)$ is initialized based on the recent arm performance estimation. Note that up to now, it is obvious that both the bias and arm performance are far from an ideal initial

Table 11: Experimental results under additional baselines.

| | Weak Regret (↓) | | | | | | Average Regret (↓) | | | | | |
|---|---|---|---|---|---|---|---|---|---|---|---|---|
| | Arm Heter. $\sigma^2$ | | | Bias Concentr. $\alpha_B$ | | | Arm Heter. $\sigma^2$ | | | Bias Concentr. $\alpha_B$ | | |
| Doubler | 791 | 815 | 889 | 1743 | 815 | 432 | 1612 | 1655 | 1693 | 2683 | 1655 | 885 |
| MultiSBM | 865 | 958 | 1045 | 1735 | 958 | 475 | 1710 | 1908 | 2055 | 2747 | 1908 | 1089 |
| MaxInP | 1394 | 1600 | 1617 | 2006 | 1600 | 711 | 2267 | 2539 | 2541 | 2982 | 2538 | 1289 |
| MaxMinLCB | 501 | 494 | 501 | 1613 | 494 | 220 | 1193 | 1227 | 1210 | 2628 | 1227 | 668 |
| BS-K(*) | **116** | **90** | **79** | **60** | **90** | 82 | **654** | 713 | 407 | **554** | 713 | **624** |
| BS-UN(*) | 194 | 161 | 94 | 340 | 161 | 92 | 690 | **689** | **387** | 825 | **689** | 637 |

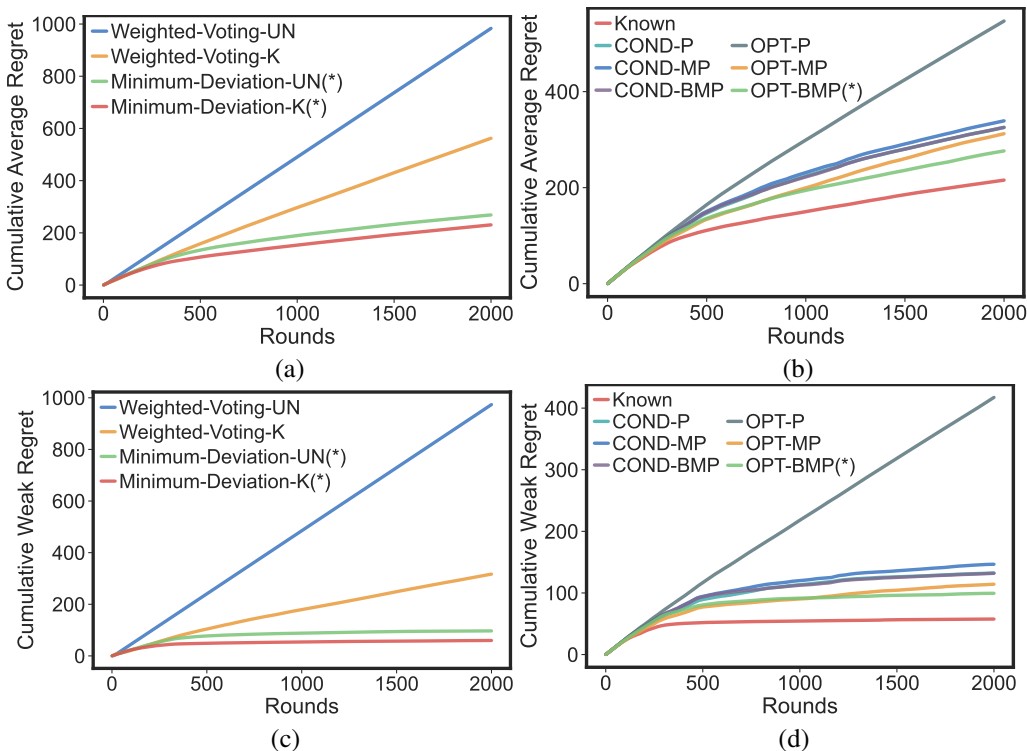

Figure 3: Ablation study: average regret of estimators on (a) arm performance and (b) bias; weak regret of estimators on (c) arm performance and (d) bias.

point for estimation update, because they are determined by ignoring the bias. To address, in line 4, we update $\hat{p}_{ij}(t-1)$ and $\hat{\eta}_m(t-1)$ for the first time. This step leads to a better initial point (for estimation update) by considering the bias. After that, the second update of and corresponds to the actual updates of the estimation.

### K.8 Regret Increase under Biased Evaluators

Table 13 shows the algorithm performance without and with evaluators' bias, motivating this work on addressing biased evaluators. We have the following observations. (i) With the presence of evaluators' bias, algorithms experience obvious increase of average and weak regrets, leading to the necessity of proposing methods to address the evaluators' bias. (ii) When there are no evaluators' bias, our proposed methods do not induce huge performance degradation, i.e., the average and weak regrets of our proposed methods remain relatively low. (iii) When evaluators' bias exist, our proposed methods achieve lower average and weak regret than most of the baselines.

Table 12: Ablation Study: Different Number of Estimation Updates.

| | Cum. Weak Regret ($\downarrow$) | Cum. Average Regret ($\downarrow$) | Rate of Best Arm ($\uparrow$) |
|---|---|---|---|
| Without Line 2 | 3752 | 6311 | 0.79 |
| Update once | 2060 | 5416 | 1.00 |
| Update twice | **1904** | **5339** | **1.00** |
| Update 3 times | 2021 | 5371 | **1.00** |
| Update 4 times | 1983 | 5369 | **1.00** |

| | Cumulative Weak Regret ($\downarrow$) | | | Cumulative Average Regret ($\downarrow$) | | |
|---|---|---|---|---|---|---|
| | No Bias | Beta$(2,1)$ | Increase | No Bias | Beta$(2,1)$ | Increase |
| RC | $122_{\pm 1.39\%}$ | $525_{\pm 2.46\%}$ | $(\times 4.31)$ | $232_{\pm 1.13\%}$ | $1338_{\pm 1.39\%}$ | $(\times 5.78)$ |
| RUCB | $154_{\pm 0.83\%}$ | $1144_{\pm 1.10\%}$ | $(\times 7.43)$ | $582_{\pm 0.34\%}$ | $2134_{\pm 0.70\%}$ | $(\times 3.67)$ |
| DT | $32_{\pm 4.65\%}$ | $492_{\pm 5.45\%}$ | $(\times 15.19)$ | $\underline{148}_{\pm 2.04\%}$ | $1425_{\pm 2.59\%}$ | $(\times 9.61)$ |
| MBTW | $\mathbf{11}_{\pm 15.47\%}$ | $162_{\pm 12.82\%}$ | $(\times 14.15)$ | $1524_{\pm 0.71\%}$ | $1539_{\pm 1.56\%}$ | $(\times 1.01)$ |
| UCB | $\underline{13}_{\pm 5.11\%}$ | $548_{\pm 3.35\%}$ | $(\times 42.26)$ | $401_{\pm 0.91\%}$ | $1426_{\pm 2.46\%}$ | $(\times 3.56)$ |
| **RC-B**(*) | $171_{\pm 1.32\%}$ | $869_{\pm 1.45\%}$ | $(\times 5.09)$ | $264_{\pm 1.18\%}$ | $1611_{\pm 1.63\%}$ | $(\times 6.10)$ |
| **RUCB-B**(*) | $361_{\pm 0.61\%}$ | $480_{\pm 0.65\%}$ | $(\times 1.33)$ | $887_{\pm 0.36\%}$ | $1120_{\pm 0.47\%}$ | $(\times 1.26)$ |
| **DT-B**(*) | $36_{\pm 18.02\%}$ | $210_{\pm 7.55\%}$ | $(\times 5.78)$ | $\mathbf{126}_{\pm 5.44\%}$ | $\mathbf{411}_{\pm 4.30\%}$ | $(\times 3.27)$ |
| **BS-UN**(*) | $\underline{31}_{\pm 2.86\%}$ | $\underline{161}_{\pm 24.46\%}$ | $(\times 5.24)$ | $447_{\pm 0.77\%}$ | $\underline{689}_{\pm 5.98\%}$ | $(\times 1.54)$ |
| **BS-K**(*) | $41_{\pm 2.44\%}$ | $\mathbf{90}_{\pm 14.02\%}$ | $(\times 2.22)$ | $458_{\pm 0.40\%}$ | $\underline{713}_{\pm 4.66\%}$ | $(\times 1.56)$ |

Table 13: Regret increase under the presence of evaluators' bias. **The methods marked with "(*)" are our methods.** The best, second, and third best results are marked in bold text, underline, and dashed underline, respectively. All these experiments are conducted under unknown bias case, expect for those of BS-K.

## Appendix L    Society Impact

This work has positive society impact on improving the dueling bandits algorithm performance in the presence of biased evaluators. For negative society impact, it may enable the agent (e.g., a platform) to detect the bias of evaluators, leading to certain privacy leakage. To address this, the proposed algorithm may be packed into package and restrict the access of the agent to the bias estimation result.

