# OpenReview forum: "Tackling Biased Evaluators in Dueling Bandits"
_NeurIPS.cc/2025/Conference — NeurIPS 2025 spotlight_

### Official Review · Reviewer_TcVQ · 2025-06-29

**Clarity:** 3
**Significance:** 2
**Originality:** 3
**Rating:** 5
**Confidence:** 4

**Summary:**

This paper addresses the previously underexplored problem of biased evaluators in the dueling bandits setting, where comparisons between arms may be corrupted. The authors first consider a benchmark setting where evaluator biases are known, and propose a bias-sensitive UCB algorithm with a novel unbiased estimator. They then extend this to the more realistic setting where the bias is unknown, proposing an iterative estimator. The paper provides theoretical regret bounds in both cases, and demonstrates that the latter has the same order of bound on the regret as the former. Extensive experiments demonstrate that the proposed methods significantly outperform standard dueling bandit algorithms, especially when evaluator bias is heterogeneous. The proposed techniques are general and can be incorporated into existing dueling bandit frameworks.

**Questions:**

As noted earlier, my primary concern and the reason for the current score is related to the regret bounds. I would appreciate clarification on the following points:
1. Based on your analysis and the observed behavior of cumulative regret over time in the experiments, do you believe that the linear dependence on $T$ in the regret upper bounds of Theorems 1 and 2 is fundamentally unavoidable when multiple evaluators are involved? If so, could you provide some theoretical or empirical evidence to support this? Conversely, if this dependence might be removable, what do you see as the main technical obstacle? Do you have any ideas or directions for how this dependence could potentially be mitigated?
2. What is the justification for assuming the existence of a ground-truth preference matrix $p_{ij}$? For instance, in the context of LLM selection—where feedback is inherently subjective—what would this ground truth represent?
3. Could you elaborate further—beyond the brief explanation in lines 71–72—on how you address the challenges involved in deriving the regret bounds?
4. You mention that your proposed estimators can be readily incorporated into existing algorithms to improve their robustness under biased evaluation. How challenging do you anticipate it would be to derive regret bounds for these modified algorithms? Can the techniques used to prove Theorems 1 and 2 be adapted to this setting, or would each algorithm require a distinct and potentially more complex analysis?
5. In Lemma 3, you derive the true confidence radius for the unknown bias setting, which involves the parameter $\eta_m$ and thus cannot be computed exactly by Algorithm 2. However, why not substitute $\eta_m$ with its estimate $\hat{\eta}_m$ in this expression, rather than relying on the less precise approximation $\hat{r}$ used in Algorithm 2? Could this substitution lead to improved empirical performance or tighter regret bounds?

**Ethical Concerns:**

["NO or VERY MINOR ethics concerns only"]

**Final Justification:**

This is the same as my comment to the authors below:

one of my principal concerns was the linear dependence on $T$ in the regret upper bounds of Theorems 1 and 2. I was pleased to see that the authors have managed to eliminate this term from Theorem 1. Regarding my other concerns—the apparent linear $T$ dependence through $\xi(t)$ in Theorem 2 and the lack of proof schemes for alternative baselines—I accept the authors’ statement that these are challenging issues that can be addressed in future work. I also acknowledge the rationale for the existence of an inherent performance matrix in this setting.

**Limitations:**

Yes

**Paper Formatting Concerns:**

There are no major formatting issues.

**Quality:**

3

**Strengths And Weaknesses:**

**Strengths:**
- **Quality:** The paper’s main claims are supported by rigorous and well-structured proofs, demonstrating a solid theoretical foundation. Additionally, the paper includes a thorough experimental evaluation that demonstrates the superiority of the proposed methods over existing baselines and provides insight into the behavior of the key parameters introduced throughout the study.
- **Clarity:** The authors clearly motivate the study of biased evaluators through a compelling example involving LLM selection. The proposed algorithms and accompanying proof techniques are presented with intuitive explanations in the main text, and their effectiveness is validated through a comprehensive and rigorous experimental evaluation.
- **Significance:** This work advances the state of the art by introducing techniques that enable effective learning in environments with biased feedback. Beyond proposing novel algorithms, the authors demonstrate that their methods can be integrated with existing approaches, which enhances the significance of this work.
- **Originality:** This is the first work to address the dueling bandits problem under biased feedback. Although the authors build on established convex optimization techniques, they employ them in a novel manner to construct provably unbiased estimators with theoretical regret guarantees.

**Weaknesses:**
Rather than separating into broad quality, clarity, significance and originality categories, I will outline my main concerns in a more detailed manner below.

- The primary concern with this work lies in the significance of the regret bounds presented in Theorems 1 and 2. As noted in Corollary 1, when the bias can take on multiple values (i.e., $|\mathcal{B}|>1$), the cumulative regret becomes linear in $T$ which is no better than the regret of a naive algorithm that selects two arms at random in each round. This issue is further exacerbated in Theorem 2, where an additional error term of the form $O(\sum^{T} \xi(t))$ appears. Unfortunately, the authors do not provide a theoretical bound for this term; instead, they offer only empirical evidence suggesting that it converges to a small value $T\epsilon$, which remains linear in $T$. As a result, this limitation substantially weakens the theoretical contribution of the paper.
- This concern is further amplified by the absence of a lower bound that would justify the observed linear cumulative regret in the presence of multiple biased evaluators. Moreover, the experimental results do not clearly indicate whether the cumulative regret grows sublinearly with the number of rounds. This raises the question of whether the linear regret is fundamentally unavoidable in this setting, or if a more refined analysis would yield improved bounds. Without addressing this, the theoretical contribution remains incomplete.
- The authors motivate the biased evaluators setting with an example involving a company acting as the agent, multiple LLMs as arms, and users as evaluators. However, the framework assumes the existence of a ground-truth pairwise preference matrix $p_{ij}$, which is not clearly justified in this context. Given that preferences over LLM outputs are inherently subjective and user-dependent, it is unclear why such a universal ground-truth should exist. This raises questions about the practical usability of this model.
- In lines 143–152, the authors state that their analysis covers both average and weak regret. However, since weak regret is always upper bounded by average regret, it seems the provided bounds effectively only analyze the latter. As a result, the regret bound, while technically applying to both, may be loose for the weak regret. Notably, in the non-biased dueling bandits setting, it is known that cumulative weak regret can be independent of $T$ [1], whereas the bound presented here remains $O(\sqrt{T})$, even under similar conditions. This observation is further supported by the experimental results, which show that weak regret is significantly lower than average regret.
- Returning to the regret upper bound in Theorem 2, the inclusion of the parameter $\xi(t)$ renders the bound somewhat incomplete, as $\xi(t)$ lacks a closed-form expression or meaningful interpretation in simple terms. Moreover, the authors do not provide a theoretical guarantee that the resulting term $O(\sum^{T} \xi(t))$ is bounded, relying instead on empirical evidence presented in Appendix I.

Other concerns –
- For a given evaluator $m$, the bias parameter $\eta_{m}$ in Equation (3) is assumed to be uniform across all arm pairs.
- In Appendix J.1, the authors describe how their techniques can be integrated into existing dueling bandit algorithms to handle biased evaluators and verify the effectiveness of this approach through experiments. However, they do not provide any accompanying theoretical analysis to support these extensions.
- The regret upper bounds in Theorems 1 and 2 are lengthy, involve numerous parameters, and are difficult to interpret. It would be preferable to move the full expressions to the appendix and present a simplified or more interpretable version in the main text.
- Additional proofreading would improve the paper. For example, “pairs” should be used instead of “pair” in line 234. The notations $p_{ij}(t)$ and $\eta_m(t)$, as used in line 240 and elsewhere, should be without $t$. In line 253, “update” should replace “updates.” In line 264, the word “for” should be removed. The definition of HF in line 281 is unclear. Line 287 appears to refer to “Assumption 3,” but likely means “Definition 3.” Finally, the parameter $t_0$ in Equation (21) seems to be used without being defined in the main paper.

[1] Saad, El Mehdi, et al. "On weak regret analysis for dueling bandits." Advances in Neural Information Processing Systems 37 (2024): 40752-40791.

---

> ### Author Rebuttal · Authors · 2025-07-30
>
> Thank you for your suggestion and identifying this work as one that is the "first work to address the dueling bandits problem under biased feedback", with "solid theoretical foundation", and "comprehensive and rigorous experimental evaluation".
>
> **Regret Bound Analysis:**
> We appreciate your valuable comments on the regret bound and agree with your statements. To address this comment, we have tried our best to improve the bound. **The improved bound is sublinear for all $B \triangleq |\mathcal{B}|$'s**. The main improvement is on inequality (f) in (42).
> -  Original Version: We bound $Z(t)$ using inequality (f) by enumerating all possible cases (i.e., all possible values of $n_{ij}^b(t)\triangleq N^b_{ij}(t)+N_{ji}^b(t)$) and relaxing every  term (each corresponding to a case) by an upper bound of one. This relaxation essentially considers each case being upper-bounded by the same value and occurring with equal chance in $\mathbb{E}[\sum_{t}Z(t)]$ in (44). However, based on our proposed algorithm, some sets of $n_{ij}^b(t)$ may occur with a very small chance. For example, for a specific $(i,j)$ pair, consider the case where the values of $n_{ij}^b(t)$ for four different $b$'s are 1, 1, 1, and 1000, respectively. This case rarely happens in our proposed algorithm, because the algorithm tends to explore arm pairs independently for evaluators with different bias levels, making the exploration times rarely highly skewed.
> - Improved Version: We provide a better bound of $\mathbb{E}[\sum_{t}Z(t)]$ by considering the different probabilities that cases occur. Specifically, we introduce an event:
> $$
> \text{Event~} X_{ij}(N_{ij},t) :  \text{There exists } b \in \mathcal{B} \text{ such that } n_{ij}^{b} - \frac{N_{ij}}{B} < - \sqrt{ \frac{N_{ij}\log(B t^{2 \alpha})}{2} } \text{ and } \sum_{b \in \mathcal{B}} n_{ij}^{b} = N_{ij}.
> $$
> Intuitively,  this is the event where the exploration is highly skewed among evaluators, as the previous example suggests.
> Then,  we decouple $Z(t)$ into two terms: $Z_1(t)$, containing terms of cases where event $X_{ij}(N_{ij},t)$ holds; $Z_2(t)$, containing terms  of cases where event $X_{ij}(N_{ij},t)$ does not hold. We can prove that (i) the probability of event $X_{ij}(N_{ij},t)$ is small, and (ii) each terms in $Z_2(t)$ is upper-bounded by a value that is much smaller than one if the associated pair $(i,j)$ has been explored for more than $m(t)=B^2\log(Bt^{2\alpha})$ times. As a result, when we take the expectation in (44), $\mathbb{E}[\sum_{t}Z_1(t)]$ is small due to the small probability, and $\mathbb{E}[\sum_{t}Z_2(t)]$ is upper bounded by a smaller bound of $\mathcal{O}(\sqrt{BT})$, because majority of the terms are upper bounded by a value smaller than one.
>
> Based on the improvement, the upper bound for $\mathbb{E}[\sum_{t}Z(t)]$ changes from $\mathcal{O}(T)$ in the original version to $\mathcal{O}(\sqrt{BT})$. Note that $B$ is the number of available bias; if the bias space is continuous (i.e., $b\in[0,1]$), $B$ can be regarded as the number of intervals for discretizing the bias space. So $B$  is usually much smaller than $T$, so the bound becomes tighter. Also, $B$ is a constant value independent of $T$.  Thus, the round-average regret improves from  $\mathcal{O}(\sqrt{\log{T}})$ to $\mathcal{O}(\sqrt{B\log{T}/T})$ and hence sublinear.
>
> For the experimental results, we have shown the convergence of the regrets in Figure 1 of Appendix J.2. The regret of our proposed algorithm appears to be sublinear.
>
> To address your comments, we will include this improved version in the manuscript. Despite that, we would like to clarify that although the recent bound analysis in the manuscript can be improved as above, this is **the first work** towards understanding the biased evaluators in dueling bandits and makes contributions in many of the other aspects:
> - We identify the **novel problem** of biased evaluators in dueling bandits.
> - We propose a dueling bandits **algorithm for known bias case**, which is *not* a simple transformation but requires careful design. We provide the **first bound** of the regret under the known bias case, overcoming challenges of heterogeneous bias level and complex formulation of the estimators.
> - We overcome the challenge of **non-convex problem** of the joint bias and winning probability estimation and propose a  block coordinate descent (BCD)-based approach for unknown bias case to solve the joint estimation problem.
> - Our approach can be **incorporated into existing** dueling bandits approach, e.g., RC, RUCB, DT, to further improve their robustness under biased evaluators. Experiment results validate the our proposed approaches **outperform** the baselines in average and weak regrets.
>
>
> **Weak Regret and Average Regret:**
> We agree with your comment. In the theoretical analysis, we essentially prove the bound on average regret. For the weak regret, we relax it to be an average regret in (33) of the proof, so the bound may be loose. To avoid over-claiming our contribution on proving the bound of weak regret, we will clearly state this in the manuscript.
>
>
> **The Inclusion of $\xi(t)$ for Unknown Bias Case:**
> We acknowledge that it will be helpful to provide the specific form of $\xi(t)$ in the bound analysis. However, this is very challenging, and we admit not being able to handle it at this moment. This is because the joint bias and winning probability estimation in (17) is a non-convex problem. Analyzing the optimality of solving a non-convex problem using BCD algorithm is an open problem, even in the area of mathematics. This is the main limitation of this work, and we will state it clearly in the manuscript. Despite this limitation, we would like to remind that this work has made substantial contributions as discussed above. We hope that this work can motivate further studies on algorithm design and analysis in the future.
>
>
> **Existence of a Ground-Truth Pairwise Preference Matrix:**
> We would like to clarify that it is a **performance** matrix but not a preference matrix. "Performance" emphasizes on the intrinsic quality of the LLM, while "preference" corresponds to user's own opinion. We use the pairwise performance matrix to characterize the ground-truth performance of the LLMs, i.e., the probability that an LLM is better than the other LLM considering all possible queries. Although this ground-truth may not be observed in practical systems, it can be defined as the average pairwise performance of LLMs across query space. We agree with the reviewer that "the LLM outputs are inherently subjective and user-dependent", i.e., **the preference can deviate from the pairwise performance**. This is the reason that motivates the consideration of user bias (i.e., user-specific preference) in the system model, e.g., some users are not sensitive to the outputs of diverse LLMs. We will clarify this point in the manuscript.
>
> **Uniform Bias:**
> Thank you for pointing out this. As the first attempt, this work considers a uniform bias for a given evaluator across all arm pairs. For example, consider a scenario where the evaluators belong to perfect evaluators, spammers (who randomize among choices), or attackers (who always choose the opposite result). In this case, the bias is the intrinsic feature of evaluators regardless of the arm pairs. We will mention the potential extension of arm-dependent bias modeling in the manuscript.
>
>
> **Elaboration on Addressing the Challenge of Bound Analysis:**
> The main difficulty comes from the complex form of the arm performance estimation and confidence radius. In particular, those are in the form of weighted sum of the feedback statistics from diverse biased evaluators. To address this, we prove new inequalities to support the bound analysis. Inequality (g) in (42) is the most important one. In this part of analysis, we transform the problem of finding an upper bound of the expression before (g) as finding the upper bound of the optimization problem given in (43), based on which we determine the upper bound of the optimization problem by analyzing the optimal solution of it in Appendix E.2.
>
>
> **Analysis on the Incorporation of Bias Estimation to Baselines:**
> Since most of the proofs in dueling bandits (e.g., UCB-based, Bayesian-based) involve the determination of the confidence radius and build upon the probability that the estimation falls within and outside the radius, we would expect that the related proof techniques on confidence radius can be applied to prove the regret bound of those baselines. However, additional techniques for accounting the specific arm performance estimation and selection approach will be needed. Take double Thompson sampling approach as an example. The arm is chosen using sampling rather than a deterministic approach, so additional stochastic behavior related to the sampling needs to be incorporated. Overall, we believe that the recent proof can help with their regret bound analysis, but further efforts are needed, which is out of scope of this paper.
>
>
> **Confidence Radius in Lemma 3:**
> If we substitute $\eta_m$ with its estimate $\hat{\eta}_m$ into the expression in Lemma 3, then by rearranging the terms, the confidence radius in Lemma 3 is exactly the same as the confidence radius in (19) used in Algorithm 2. Note that this does not imply that if we substitute $\hat{\eta}_m$ with $\eta_m$ into (19), then we can directly obtain the actual confidence radius in (20). This is because the confidence radius corresponds to the radius where an estimation of arm performance falls within the radius (centered around the actual arm performance) with a sufficiently high probability, so the actual confidence radius is a function of both the actual bias $\eta_m$ and the estimated one $\hat \eta_m$.
>
> **Typos and Formatting:**
> We will address them in the manuscript.

---

> > ### Comment · Reviewer_TcVQ · 2025-08-03
> >
> > I thank the authors for their detailed response and efforts.
> >
> > One of my principal concerns was the linear dependence on $T$ in the regret upper bounds of Theorems 1 and 2. I was pleased to see that the authors have managed to eliminate this term from Theorem 1. Regarding my other concerns—the apparent linear $T$ dependence through $\xi(t)$ in Theorem 2 and the lack of proof schemes for alternative baselines—I accept the authors’ statement that these are challenging issues that can be addressed in future work. I also acknowledge the rationale for the existence of an inherent performance matrix in this setting.
> >
> > For these reasons, I have raised my score. I recommend that the authors expand upon the points raised during the rebuttal in their revised version, and I wish them success with the paper.

---

> > > ### Author Response · Authors · 2025-08-04
> > >
> > > Thank you for your valuable feedback and for raising your score. We are glad that the revisions to Theorem 1 addressed your concern regarding the linear dependence on $T$, and appreciate your understanding regarding the challenges associated with Theorem 2. We will expand upon these points in the revised manuscript as suggested.
> > >
> > > Thank you again for your time and insightful comments.

---

### Official Review · Reviewer_dKuD · 2025-06-30

**Clarity:** 4
**Significance:** 3
**Originality:** 3
**Rating:** 5
**Confidence:** 4

**Summary:**

The authors consider the duelling bandit problem with finite arms and biased evaluators.
In particular, they assume that there is a finite set of evaluators that are either provide feedback according to the true preferences or the contrary in pairwise comparisons. Each evaluator provides good feedback with certain probability (their bias variable).
In the first part, the authors derive a no-regret algorithm for the case when the evaluators' biases are known.
In the second part, they extend the case where both the preferences and the biases are unknown.
Finally, the authors provide experimental evidence that their proposed algorithms outperform algorithms that disregard evaluator biases and their confidence estimation can be incorporated in many other algorithms to improve their performance.

**Questions:**

1. Why do the authors choose the Borda Score as their notion of optimality in the general preference setting? I would encourage them to discuss strengths and weaknesses of this design choice. E.g. how does it represent minorities in the annotator pool?
2. In Line 222-223, the authors claim "This sublinearity result is consistent with and generalizes those existing works on dueling bandits
without considering evaluators’ bias (e.g., [13])" could they provide an elaboration on this claim?
3. The authors use the term "highly coupled" multiple times. Could they specify what they mean exactly? I found this fairly ambiguous.
4. Line 264-265, the authors claim "We empirically show that performing such updates twice leads to the best performance" but I could not find these empirical evaluations. Could you point me to them?
5. Could the authors also report the cumulative regret curves in the Experiments section corresponding to Table 1 and 2 instead of just reporting the values at a specific time-step?
6. Could the authors elaborate why they choose RC, RUCB, DT, and MBTW as comparison algorithms and omit other works such as Double, MultiSBM [Ailon et al. 2014], MaxInP [Saha, 2021] and MaxMinLCB [Pasztor et al. 2024] that show good performance on the standard duelling bandit problem?

**Ethical Concerns:**

["NO or VERY MINOR ethics concerns only"]

**Final Justification:**

To my judgement, this paper has significant contribution and the authors' adequately answered all the questions I raised in my review. I maintain my positive score and support this work's acceptance.

**Limitations:**

Please see the Weaknesses section. I have listed my concerns there.

**Quality:**

3

**Strengths And Weaknesses:**

I found the following points to be strengths of this paper:
1. General preference case not restricted to the Bradley-Terry or Logistic case. This means it is applicable to complex preferences that might not be transitive or follow a reward model.
2. The results on estimating the pairwise comparison probabilities and biases are general and can be incorporated in many other algorithms
3. The investigation of biases (especially in human feedback) is more and more crucial given their use in training frontier models and deployed to the general public

Weaknesses:
1. Considering finite arms and set of evaluators limit the applicability of these results to applications such as training LLMs which the authors use as a motivating example
2.  I found the motivating example of a company choosing LLMs for deployment to be a bit misleading. In natural language, biases are often contextual (e.g. depends on the question posed). Accordingly some models might be better for certain contexts than others and hard to find a single best option. Additionally, these contexts are infinite (but structured) and the proposed approach can not be easily extended to this case.
3. In the general preference model setting, the definition of optimality is non-trivial. The authors choose the Borda winner as their notion of optimality but do not discuss its advantages and disadvantages such as neglecting minorities.
4. While the experiments provide some experimental evidence its is limited to 10 arms and 10 evaluators and a single preference model. Further experiments in larger settings are required to demonstrate the applicability of their results.

---

> ### Author Rebuttal · Authors · 2025-07-30
>
> Thank you for your suggestions and for identifying the strength of this work on "general preference" setting, "general" estimation approach, and the "crucial" investigation of bias.
>
> **Finite Arms and Evaluators:**
> We would like to clarify that in the LLM example, the number of arms (i.e., LLMs) and the number of evaluators (i.e., humans) are finite in practice. Although the number of evaluators can be huge, each person frequently interacts with LLMs and provides feedback, under which it is still possible to estimate the bias of evaluators and hence estimate the arm performance.
>
> **Bias Due to Contextual Information:**
> Thank you for pointing out this interesting point. We agree with the reviewer that the performance of LLMs depends on contextual information and the available contexts can be infinite. First, this is the reason why we introduce a probability-based performance model for arms, i.e., the pairwise comparison outcome between two arms is modeled as a random variable rather than being deterministic, as most of the works on dueling bandits (Bengs et al., 2021). Second, we agree with the reviewer that incorporating contextual information for arm comparison can better characterize the scenario in the motivating example. To achieve this, one potential idea is to incorporate contextual information into the winning probability modeling (Ong et al. 2025). In this case, we need to take into account the context similarity in the arm performance and bias estimation. Overall, this work aims to investigate a general dueling bandits scenario instead of working on a specific application scenario, so as the first step, we simply the model by focusing solely on the impact of biased evaluators on dueling bandits.  We will include these discussions in the manuscript to alleviate the potential confusion.
>
> I. Ong, A. Almahairi, V. Wu, W. Chiang, T. Wu, J. E. Gonzalez, M. W. Kadous, and I. Stoica, " RouteLLM: Learning to Route LLMs from Preference Data," Proc. ICLR, Singapore, April 2025.
>
> **Advantage and Disadvantage of Optimality Definition:**
> In this work, we choose to consider the Borda winner for two reasons. First, it is commonly considered in dueling bandits, as shown in Tables 6 and 7 of the survey paper (Bengs et al., 2021). Second, it reveals the overall probability that an arm wins over other arms. Specifically, the Borda winner is the arm with the highest winning probability. This is in contrast to, for example, ​​the Copeland winner, which is the one that beats more arms (but possibly with a winning probability only slightly higher than 0.5). Considering the motivating example of LLM inference with diverse queries, the Borda winner could be more suitable, as a higher overall winning probability likely indicates a higher chance that users are satisfied with the inference results.
>
> We agree with the reviewer that defining the optimality using the Borda winner has limitations. From the theoretical perspective, one major limitation of the Borda winner is that it may sometimes not be consistent with the Condorcet winner (if it exists). This Condorcet winner is the best arm among arms if a total ordering of arms exists. However, the Condorcet winner does not usually exist in practical systems (Bengs et al., 2021), e.g., there may not exist an LLM that is better than all other LLMs for all kinds of queries. Thus, the aforementioned limitation may not be a major concern. On the other hand, in terms of neglecting minorities in the annotator pool, each annotator's preference/feedback contributes to the winning probability of arms with an equal weight in the Borda winner. Thus, our model does not neglect or exclude minority opinions. Thank you for pointing out this valuable point. We will include these discussions in the manuscript.
>
> **Experiments with More Arms/Evaluators and Preference Model:**
> Due to the space limit, we place some of the experiments in the Appendix. We have conducted experiments with up to 100 arms and 100 evaluators. Please refer to Table 7 in Appendix J.4 for details. In addition, we have evaluated the impact of the preference model by varying the arm heterogeneity. Please refer to Tables 1, 3, and 4 for details.
>
> **Sublinearity Result Being Consistent with Existing Works:**
> With this statement, we mean that if there is no bias, then the order of the regret bound of our algorithm is the same as those in the existing works that did not consider bias, e.g., [13]. The order is with respect to $T$. Specifically, our regret is at the order of $\mathcal{O}(\sqrt{\log{T}/(T\Gamma^2)})$, where $\Gamma$ can be regarded as a constant value; in [13] which did not consider bias, its algorithm achieves a regret at order $\mathcal{O}(\sqrt{\log{T}/T})$. It can be seen that they have the same order with respect to $T$. We will include this explanation in the manuscript.
>
>
> **Explanation on Highly Coupled:**
> The "highly coupled" of the arm performance and bias estimation means that the estimation of arm performance is a function of the estimated bias in (16), and the estimation of the bias is a function of the estimated arm performance in (18). In this case, we cannot estimate them in a sequential manner, i.e., one after the other. It is also challenging to estimate them simultaneously due to the non-convexity of the estimation problem. Also, an inaccurate estimation of one of them will lead to an inaccurate estimation of the other, making the algorithm design difficult. We will include the explanation in the manuscript.
>
> **Explanation and Evaluation on "Update Twice":**
> The following shows the evaluation of different times of updates. Note that to show a relatively obvious performance gap, we conducted experiments under a larger scale setting than that in Table 1, which has 50 arms and 10 evaluators.
> |Algorithms|Cumulative Weak Regret|Cumulative Average Regret|Rate of Best Arm|
> |-------|-------|-------|-------|
> |Without Init. in Line 2|3752|6311|0.79|
> |Update once|2060|5416|**1.00**|
> |Update twice|**1904**|**5339**|**1.00**|
> |Update 3 times|2021|5371|**1.00**|
> |Update 4 times|1983|5369|**1.00**|
>
> It can be observed that "updating twice" leads to the best performance. Further increasing the times can degrade the performance. This result will be included in the Appendix. Then, we would like to explain the intuition for "updating twice". The main idea is to find a better initial point for a more accurate estimation in each iteration. According to Algorithm 2 in Appendix F, in line 2 of each iteration, the arm performance estimation $\hat p_{ij}(t-1)$ is initialized as the estimation ignoring the bias, and then the bias estimation $\hat \eta_{m}(t-1)$  is initialized based on the recent arm performance estimation $\hat p_{ij}(t-1)$. Note that up to now, it is obvious that both the bias and arm performance are far from an ideal initial point for estimation update, because they are determined by ignoring the bias. To address, in line 4, we update $\hat p_{ij}(t-1)$ and $\hat \eta_{m}(t-1)$ for the first time. This step leads to a better initial point (for estimation update) by considering the bias. After that, the second update of $\hat p_{ij}(t-1)$ and $\hat \eta_{m}(t-1)$ corresponds to the actual updates of the estimation.
>
>
> **Cumulative Regret Curve:**
> The cumulative regret curves are provided in the Appendix due to the space limit. Please see Figure 1 in Appendix J.2 for details.
>
> **Additional Baselines:** Thank you for your suggestion. We have conducted experiments on Doubler, MultiSBM, MaxInP, and MaxMinLCB. The following provides the experimental results on weak and average regrets respectively.
> |Weak Regret|$\sigma^2=1.0$|$\sigma^2=2.0$|$\sigma^2=4.0$|$\alpha_{\text{B}}=1.0$|$\alpha_{\text{B}}=2.0$|$\alpha_{\text{B}}=3.0$|
> |-------|-------|-------|-------|-------|-------|-------|
> |Doubler|791|815|889|1743|815|432|
> |MultiSBM|865|958|1045|1735|958|475|
> |MaxInP|1394|1600|1617|2006|1600|711|
> |MaxMinLCB|501|494|501|1613|494|220|
> |BS-K(*)|**116**|**90**|**79**|**60**|**90**|**82**|
> |BS-UN(*)|194|161|94|340|161|92|
>
> |Average Regret|$\sigma^2=1.0$|$\sigma^2=2.0$|$\sigma^2=4.0$|$\alpha_{\text{B}}=1.0$|$\alpha_{\text{B}}=2.0$|$\alpha_{\text{B}}=3.0$|
> |-------|-------|-------|-------|-------|-------|-------|
> |Doubler|1612|1655|1693|2683|1655|885|
> |MultiSBM|1710|1908|2055|2747|1908|1089|
> |MaxInP|2267|2539|2541|2982|2538|1289|
> |MaxMinLCB|1193|1227|1210|2628|1227|668|
> |BS-K(*)|**654**|713|407|**554**|713|**624**|
> |BS-UN(*)|690|**689**|**387**|825|**689**|637|
>
> The results show that our proposed approaches BS-K and BS-UN outperform the baselines in both weak and average regrets. We will include them in the manuscript.

---

> > ### Comment · Reviewer_dKuD · 2025-08-04
> >
> > Thank you for addressing my questions and the additional pointers to results in the Appendix. The additional discussions in the paper and extensions to the experimental results clarified the results and claims of the work. I would like to maintain my score and evaluation that this is a solid work meeting the requirements for acceptance.

---

> > > ### Author Response · Authors · 2025-08-05
> > >
> > > Thank you for your insightful comments and for ​​maintaining your positive evaluation​. We are pleased that the clarifications and extended discussions/experiments addressed your points. Thank you for your time and support.

---

### Official Review · Reviewer_wbVs · 2025-07-18

**Clarity:** 4
**Significance:** 3
**Originality:** 3
**Rating:** 5
**Confidence:** 4

**Summary:**

The authors tackle the problem that the binary feedback in a dueling bandit scenario, e.g. given by a human evaluator, might be biased. In their setting, the bias refers to the probability that the expected feedback from a specific evaluator coincides with the ground truth pairwise winning probabilities. In the paper, two settings, with known and unknown bias, are covered. For both, they provide a closed form to estimate the pairwise winning probabilities between the arms (and resp. the bias of the evaluators) and propose UCB-based algorithm to solve the regret minimization problem, while they focus on average and weak regret. In a theoretical analysis, the authors provide regret bounds for both cases, which is in the case of unknown bias far from trivial due to the non-convexity of the problem. In an experimental study, it is shown that their methods suffer less regret in comparison to classical dueling bandit algorithms as baselines and moreover, that these baselines could be extended and improved by their approach for the bias estimation.

**Questions:**

Questions:
- I am wondering how the weak and average regret can have the same regret bounds. As the name suggests, the weak regret is usually lower than the average regret (assuming that not two times the same arm is chosen during a time step), so intuitively I would expect a tighter regret bound for the weak regret. Can you discuss how tight the bounds are? What is about the strong regret?

- Usually, in dueling bandit methods like in Relative UCB, only the first arm is chosen as the one that maximizes the upper confidence bound, and the second arm is chosen as the one that is the "best competitor" for the first chosen arm where the winning probability of the first arm is minimized. Formally, the first chosen arm would be $x_1(t) = max_i UCB_i(t)$ and the second chosen one $x_2(t) = min_j UCB_{i,j}(t)$. Have you tried out this approach or do you have a reason why you don't do it like this?

- How does your method with unknown bias behave if no bias is present? Does it correctly recognize that there is no bias? Can it keep up with the baselines like RUCB or double thompson sampling? Does the difference between your methods and the baselines become bigger with larger bias? For this, some experiments with different beta distributions to sample the bias would be nice

**Ethical Concerns:**

["NO or VERY MINOR ethics concerns only"]

**Final Justification:**

As already stated above, the paper is solid work regarding the well-motivated idea, the proven regret bounds, and the experimental study. I want to positively outline the rebuttal of the authors who were willing to make some points clearer in the final version of their paper and, in addition, provided some new, insightful experimental results.

**Limitations:**

The authors mentioned some limitations, like the assumption that the bias is fixed over the whole time horizon and the lack of experiments on real-world human data. Other potential limitations (maybe a slower convergence of the estimated pairwise probabilities if no bias is present?) are not made clear and cannot be derived by the reader from the shown experiments.

**Paper Formatting Concerns:**

No concerns.

**Quality:**

3

**Strengths And Weaknesses:**

The topic is motivated well with a recently relevant application of the output selection of LLMs. Although I did not read the proofs in the appendix in full detail, the biggest strength of the paper is that the proposed approach is theoretically grounded and provided with a regret analysis for weak as well as for average regret that generalizes over the known regret bounds for the case of no existing bias. The whole paper is well written and easily understandable. However, a weakness is the experimental study, which covers to the best of my understanding only synthetic data and only one instantiation of the distribution from which the biases are sampled. Here, a more detailed sensitivity analysis of how the methods behave, for example, for no bias at all or for different beta distributions from which the biases are sampled, would be insightful.

---

> ### Author Rebuttal · Authors · 2025-07-30
>
> Thank you for your suggestions and identifying the strength of this work on "theoretically grounded", being "motivated well", and "well-written".
>
> **Bias Distribution:**
> Thank you for pointing this out. We have conducted experiments under different bias distributions. Due to the space limit, some of the results are presented in the Appendix. Here are the details:
> - Different Beta Distributions: In the experiments, we use $\alpha_{\text{B}}$ in the beta distribution to characterize the impact of bias. A smaller $\alpha_{\text{B}}$ implies a higher degree of evaluators’ bias. Please refer to Tables 1, 3, and 4 for the results. As the reviewer expects, our proposed approaches are more beneficial when the bias is more severe. Take the weak regret in Table 1 as an example. When $\alpha_{\text{B}}$ decreases from 3.0 to 1.0, the regret reduction improves from 13.0%-83.8% to 89.5%-96.8%.
> - No Bias: Please refer to Table 8 in Appendix J.6. For the weak regret, the results of the baselines range from 11 to 154, and the best performance among our approaches is 31. For the average regret, the results of the baselines range from 148 to 1524, while the best performance among our approaches is 126, which is superior to the baselines. In summary, our proposed algorithms may lead to a performance degradation when there is no bias. However, the degradation may not be large, indicating a capability to correctly recognize the absence of bias.
>
> **Relative UCB:**
> We have conducted experiments on Relative UCB, which is denoted by "RUCB" in the manuscript.
>
> **Weak and Average Regrets:**
> Thank you for pointing this out. We agree with your statements. In the theoretical analysis, we essentially prove the bound on average regret. For the weak regret, we relax it to be an average regret in (33) of the proof, so the bound can be loose. Due to this reason, our analysis cannot be applied to strong regret. We will state this point clearly in the manuscript.
>
> **Synthetic Data:**
> This work is more likely to be a theoretical work (rather than an application work), so we use synthetic data in experiments. This provides us with a more systematic way to understand how bias distribution affects the algorithm performance. We agree with the reviewer that experiments with real-world bias data can help to validate the algorithm performance in practical systems. However, this is very challenging during the short rebuttal period, and we will include it in the future work.

---

> > ### Comment · Reviewer_wbVs · 2025-08-04
> >
> > Thank you for your answer and for clarifying some of my points and questions. The comment I raised regarding RUCB was not that I was missing experiments concluding RUCB, but the arm selection strategy in RUCB is slightly different from yours (as described in my review above). I was just wondering whether there is a reason why you chose a different selection strategy. However, since this is only a minor point and a question out of curiosity, I still claim to accept this paper and maintain my score.

---

> > > ### Author Response · Authors · 2025-08-06
> > >
> > > Thank you for this insightful feedback. We consider to choose the arms with the highest and second highest Borda scores $\theta_i$ primarily due to the analytical simplicity. In this case, the choice of the second arm is independent of the choice of the first arm, making the algorithm design and regret bound analysis simpler.
> > >
> > > In response to the reviewer's question, we conducted experiments and showed that our arm selection approach and the approach mentioned by the reviewer achieve similar performance for most of the cases, while our approach achieves better performance under large-scale setting and the scenario when a Condorcet winner does not exist.
> > >
> > > **Approaches for Comparison:** In response to the reviewer's question, we focus on comparing the choices of the second arm, i.e., either the arm with the second highest Borda score  or the "best competitor" for the first chosen arm.
> > >
> > > |  | First Arm  | Second Arm | Bias Estimation |
> > > |:--------|:--------|:--------|:--------|
> > > | RUCB-B   |  Randomly pick from an optimistic arm pool  | "Best competitor"   | Our Bias Estimation   |
> > > | BS-K-Modified  | Highest Borda score   | "Best competitor"    |  Known Bias |
> > > | BS-K | Highest Borda score  | Second highest Borda score   | Known Bias |
> > > | BS-UN-Modified | Highest Borda score   | "Best competitor"    |  Our Bias Estimation |
> > > | BS-UN | Highest Borda score  | Second highest Borda score   | Our Bias Estimation |
> > >
> > > In summary, RUCB-B is built upon Relative UCB [11] while incorporating our bias estimation. BS-K-Modified and BS-UN-Modified correspond to the arm selection approach mentioned by the reviewer, incorporated with our arm performance estimation method. BS-K and BS-UN are our approaches.
> > >
> > > **Experimental Results:**  In the following results, we pair BS-K-Modified and BS-K and pair BS-UN-Modified and BS-UN, and use the bold text to indicate the better result within each pair. Each column shows a different number of arms.
> > >
> > > Scenario 1: Bradley-Terry (BT) model
> > >
> > > The arm performance matrix is generated using the BT model, same to the setting in the manuscript.
> > >
> > > |Weak Regret |10|20|30|50|
> > > |:--------|:--------|:--------|:--------|:--------|
> > > |RUCB-B|685 |1444 |2451 |6468|
> > > ||||||
> > > |BS-Modified-K|59 |**322** |644 |1732|
> > > |BS-K|**57** |336 |**537** |**1256**|
> > > ||||||
> > > |BS-Modified-UN|96 |**418** |815 |2291 |
> > > |BS-UN|**92** |477 |**687** |**1355**|
> > >
> > > |Average Regret |10|20|30|50|
> > > |:--------|:--------|:--------|:--------|:--------|
> > > |RUCB-B|1943 |2914 |4280 |8990 |
> > > ||||||
> > > |BS-Modified-K|**756** |**1308** |**1592** |5155 |
> > > |BS-K|868 |1361 |1748 |**4440**|
> > > ||||||
> > > |BS-Modified-UN|**829** |1469 |1951 |5643 |
> > > |BS-UN|990 |**1461** |**1834** |**4109** |
> > >
> > >
> > > Scenario 2: Non-Existence of a Condorcet Winner
> > >
> > > The arm performance matrix is initialized with the BT model. To remove the Condorcet winner, for each arm, we randomly select two arms that are initially weaker than this arm and  increase their winning probabilities (that beat this arm) to a random value within (0.5, 0.6).
> > >
> > > |Weak Regret |10|20|30|50|
> > > |:--------|:--------|:--------|:--------|:--------|
> > > |RUCB-B|1133 |1913 |2841 |6238 |
> > > ||||||
> > > |BS-Modified-K|224 |**476** |865 |2297 |
> > > |BS-K|**198** |588 |**755** |**1631** |
> > > ||||||
> > > |BS-Modified-UN|**276** |811 |990 |2661 |
> > > |BS-UN|437 |**697** |**907** |**1960** |
> > >
> > > |Average Regret |10|20|30|50|
> > > |:--------|:--------|:--------|:--------|:--------|
> > > |RUCB-B|3619 |3966 |4918 |8619 |
> > > ||||||
> > > |BS-Modified-K|2602 |2387 |2625 |5205 |
> > > |BS-K|**1715** |**1787** |**2146** |**4737** |
> > > ||||||
> > > |BS-Modified-UN|2742 |2701 |2808 |5418 |
> > > |BS-UN|**1401** |**1768** |**2186** |**4648** |
> > >
> > >
> > > Main observations:
> > > 1.  For most of the cases, BS-K-Modified and BS-K (as well as BS-UN-Modified and BS-UN) achieve similar weak and average regrets. This indicates that those two methods for choosing the second arm do not make significance difference.
> > > 2. When there are 50 arms,  our BS-K and BS-UN always outperform BS-K-Modified and BS-UN-Modified, respectively. This indicates that independently selecting two arms (rather than the "best competitor") for exploration is more beneficial for arm performance estimation when the number of arms is large.
> > > 3. When the Condorect winner does not exist, our BS-K and BS-UN always outperform BS-K-Modified and BS-UN-Modified in terms of the average regret, respectively. In this case, the “best competitor" for the first chosen arm may perform badly when compared with other arms, so selecting the “best competitor" as the second arm may lead to a high regret and hence increase the average rerget.
> > > 4. RUCB-B achieves the worst performance. This result shows that the choice of the first arm makes more significant impact than that of the second arm.
> > >
> > > Finally, it is challenging to conduct a comprehensive evaluation during this limited rebuttal period. We hope that this work, your insightful comment, and the above discussions can motivate further investigation on this interesting point.
> > >
> > > Thank you again for your time and insightful feedback.

---

### Decision · Program_Chairs · 2025-09-17

**Decision:**

Accept (spotlight)

**Comment:**

The authors address the issue of bias in binary feedback in a dueling bandit scenario. In their model, bias is defined as the probability that the expected feedback from a specific evaluator will coincide with the true pairwise winning probabilities. The paper covers two settings, with known and unknown bias. They provide a closed-form expression for estimating the pairwise winning probabilities between arms for both. They also propose a UCB-based algorithm for solving the problem of minimising regret, focusing on average and weak regret. In their theoretical analysis, the authors provide regret bounds for both cases. An experimental study shows that their methods incur less regret than classical dueling bandit algorithms.

Overall, the reviewers are quite positive about the paper. In their reports, they raised a couple of comments and concerns, however, these could essentially be resolved in the rebuttal and subsequent discussion.